# GravMAD: Grounded Spatial Value Maps Guided Action Diffusion for Generalized 3D Manipulation

Yangtao Chen[*1], Zixuan Chen[*1], Junhui Yin[1], Jing Huo[†1], Pinzhuo Tian[3], Jieqi Shi[2], Yang Gao[1,2]

[1]State Key Laboratory for Novel Software Technology, Nanjing University, Nanjing, China
`yangtaochen@smali.nju.edu.cn`, `{chenzx,huojing,gaoy}@nju.edu.cn`,
`yinjunhui@smail.nju.edu.cn`
[2]School of Intelligence Science and Technology, Nanjing University (Suzhou Campus), Suzhou, China
`jayceesjq@gmail.com`
[3]School of Computer Engineering and Science, Shanghai University, Shanghai, China
`pinzhuo@shu.edu.cn`

## Abstract

Robots' ability to follow language instructions and execute diverse 3D manipulation tasks is vital in robot learning. Traditional imitation learning-based methods perform well on seen tasks but struggle with novel, unseen ones due to variability. Recent approaches leverage large foundation models to assist in understanding novel tasks, thereby mitigating this issue. However, these methods lack a task-specific learning process, which is essential for an accurate understanding of 3D environments, often leading to execution failures. In this paper, we introduce **Grav-MAD**, a sub-goal-driven, language-conditioned action diffusion framework that combines the strengths of imitation learning and foundation models. Our approach breaks tasks into sub-goals based on language instructions, allowing auxiliary guidance during both training and inference. During training, we introduce **Sub-goal Keypose Discovery** to identify key sub-goals from demonstrations. Inference differs from training, as there are no demonstrations available, so we use pre-trained foundation models to bridge the gap and identify sub-goals for the current task. In both phases, **GravMaps** are generated from sub-goals, providing GravMAD with more flexible 3D spatial guidance compared to fixed 3D positions. Empirical evaluations on RLBench show that GravMAD significantly outperforms state-of-the-art methods, with a 28.63% improvement on novel tasks and a 13.36% gain on tasks encountered during training. Evaluations on real-world robotic tasks further show that GravMAD can reason about real-world tasks, associate them with relevant visual information, and generalize to novel tasks. These results demonstrate Grav-MAD's strong multi-task learning and generalization in 3D manipulation. Video demonstrations are available at: `https://gravmad.github.io`.

## 1 Introduction

One of the ultimate goals of general-purpose robot manipulation learning is to enable robots to perform a wide range of tasks in real-world 3D environments based on natural language instructions (Hu et al., 2023a). To achieve this, robots must understand task language instructions and align them with the spatial properties of relevant objects in the scene. Additionally, robots must effectively generalize across different tasks and environments; otherwise, their practical application will be limited (Zhou et al., 2023). For example, if a robot has learned the policy for the task *"Take the chicken off the grill"*, it should also be able to perform the task *"Put the chicken on the grill"*. Without this generalization ability, its utility will be greatly reduced. Recent research in robot learning for 3D manipulation tasks has focused on two mainstream approaches: imitation learning-based methods and pre-trained foundation model-based methods. Imitation learning-based methods learn end-to-end policies from

---

[*]These authors contributed equally.
[†]Corresponding author.

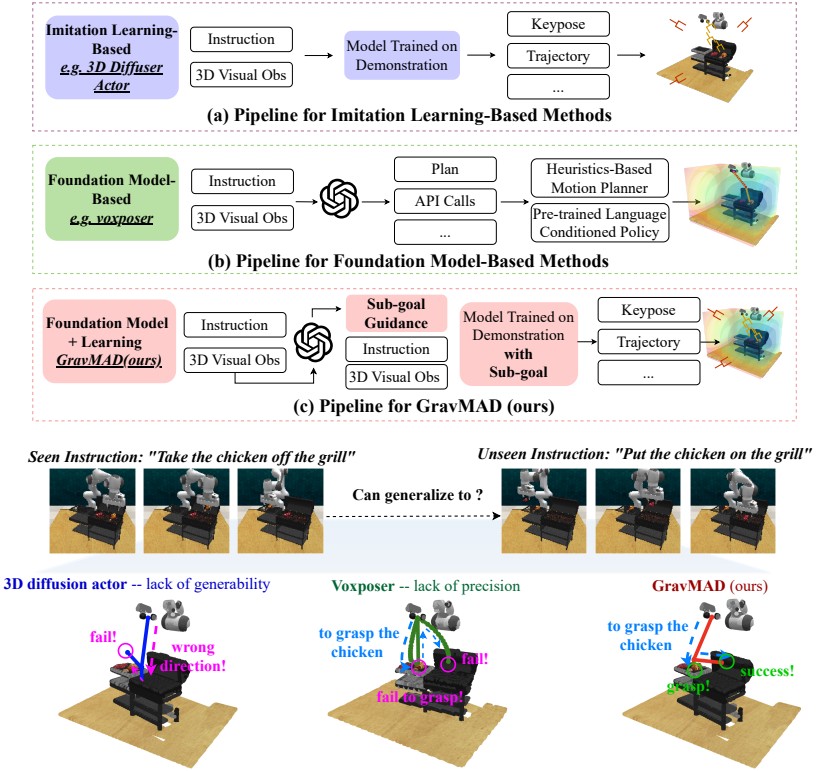

(a) Pipeline for Imitation Learning-Based Methods

(b) Pipeline for Foundation Model-Based Methods

(c) Pipeline for GravMAD (ours)

(d) Comparison of Generalization Capabilities of the Three Types of Methods

Figure 1: **Comparison of Pipelines.** (a) Imitation learning-based methods learn end-to-end policies that map language and 3D observations to actions for precise manipulation. (b) Foundation models-based methods use LLMs/VLMs to process inputs, generate plans, and execute actions with predefined primitives for task generalization. (c)(d) GravMAD combines both, using sub-goal guidance to leverage the language understanding of foundation models and the policy learning of imitation learning for precise and generalized manipulation.

expert demonstrations in attempt to address 3D manipulation tasks (Walke et al., 2023; Padalkar et al., 2024; Argall et al., 2009; Chen et al., 2024a). By designing various learning frameworks, such as incorporating different 3D representations (Shridhar et al., 2023; Chen et al., 2023a; Goyal et al., 2023), policy representations (Ze et al., 2024; Ke et al., 2024; Yan et al., 2024), and multi-stage architectures (Gervet et al., 2023; Goyal et al., 2024), imitation learning-based policies can map perceptual information and language instructions to actions that complete complex 3D manipulation tasks. However, these policies often overfit to specific tasks (Xie et al., 2024; Zhang et al., 2024), leading to significant performance degradation or even failure when applied to tasks that differ from those encountered during training (Brohan et al., 2023a; Zitkovich et al., 2023).

Another line of cutting-edge research seeks to leverage foundation models trained on internet-scale data (OpenAI, 2023; Yang et al., 2023b) to enhance policy generalization across a variety of tasks (Brohan et al., 2023b; Hu et al., 2023b; Huang et al., 2023). Unlike traditional imitation learning-based methods, approaches using pre-trained foundation models typically decouple perception, reasoning, and control during manipulation (Sharan et al., 2024). However, this decoupling often leads to a limited understanding of scenes and manipulation tasks (Huang et al., 2024), allowing robots to conceptually grasp tasks but failing to accurately complete tasks in 3D environments, resulting in failures. This underscores a key challenge: both imitation learning-based and foundation model-based approaches struggle to balance precision and generalization when adapting to novel 3D manipulation tasks. Such a challenge raises a crucial question: *Can the strengths of both approaches be combined to achieve precise yet generalized 3D manipulation?*

To this end, inspired by the approach of introducing task sub-goals to achieve efficient execution in robotic manipulation (Black et al., 2024; Kang et al., 2023; Xian et al., 2023; Ma et al., 2024), we propose discovering key sub-goals for 3D manipulation tasks as a bridge between foundation models

and learned policies, leading to the development of **Gr**ounded Sp**a**tial **V**alue **M**aps-guided **A**ction **D**iffusion (**GravMAD**), a novel sub-goals-driven, language-conditioned action diffusion framework. Specifically, a new data distillation method called **Sub-goal Keypose Discovery** is introduced during the training phase. This method identifies the key sub-goals required for each sub-task stage from the demonstrations. In the inference phase, pre-trained foundation models are leveraged to interpret the robot's 3D visual observations and task language instructions, directly identifying task sub-goals. Once the task sub-goals are obtained, the voxel value maps introduced in Voxposer (Huang et al., 2023) are used to generate the corresponding **Gr**ounded Sp**a**tial **V**alue **Map**s (**GravMaps**). These maps reflect both the cost associated with each sub-goal and the ideal gripper openness. The closer to the sub-goal, the lower (cooler) the cost; the farther away, the higher (warmer) the cost, while also indicating the gripper's state within the sub-goal range. Thus, they serve as intuitive tools for grounding language instructions into 3D robotic workspaces. Finally, the generated GravMaps are integrated with the policy diffusion architecture proposed in 3D diffuser actor (Ke et al., 2024), forming the GravMAD framework. This enables the robot to utilize 3D visual observations, task language instructions, and GravMaps guidance to denoise random noise into precise end-effector poses. As shown in Fig. 1, GravMAD effectively combines the precise manipulation capabilities of imitation learning-based methods with the reasoning and generalization abilities of foundation model-based approaches. We extensively evaluate GravMAD on RLBench (James et al., 2020), a representative benchmark for instruction-following 3D manipulation tasks. The results show that GravMAD not only performs well on tasks encountered during training but also significantly outperforms state-of-the-art baseline methods in terms of generalization to novel tasks. Additionally, we validate these findings through 10 real-world robotic manipulation tasks.

In summary, our contributions are: **1)** We propose leveraging key sub-goals in 3D manipulation tasks to bridge the gap between foundation models and learned policies. In the training phase, we introduce a data distillation method, **Sub-goal Keypose Discovery**, to identify task sub-goals. In the inference phase, foundation models are used for this purpose. **2)** We generate **GravMaps** from these sub-goals, translating task language instructions into 3D spatial sub-goals and reflecting spatial relationships in the environment. **3)** We propose a new action diffusion framework, **GravMAD**, guided by GravMaps. It is sub-goal-driven and language-conditioned, combining the precision of imitation learning with the generalization capabilities of foundation models. **4)** The simulation experiments are conducted on 20 tasks in RLBench, comprising two types: 12 base tasks directly selected from RLBench, and 8 novel tasks created by modifying scene configurations or task instructions. GravMAD achieves at least **13.36%** higher success rates than state-of-the-art baselines on the 12 base tasks encountered during training, and surpasses them by **28.63%** on the 8 novel tasks, highlighting its strong generalization capabilities. Experiments on 10 real-world robotic tasks further validate GravMAD's effectiveness.

## 2 RELATED WORKS

**Learning 3D Manipulation Policies from Demonstrations.** Recent works have employed various perception methods to learn 3D manipulation policies from demonstrations to tackle the complexity of reasoning in 3D space. These methods include using 2D images (Chen et al., 2024b; Brohan et al., 2023a; Zitkovich et al., 2023; Jang et al., 2022), voxels (Shridhar et al., 2023; James et al., 2022), point clouds (Chen et al., 2023a; Yuan et al., 2023), multi-view virtual images (Chen et al., 2023b; Goyal et al., 2023; 2024), and feature fields (Gervet et al., 2023). To support policy learning, some studies (Ke et al., 2024; Xian et al., 2023; Yan et al., 2024; Ze et al., 2024) have integrated 3D scene representations with diffusion models (Ho et al., 2020). These approaches attempt to handle the multi-modality of actions, in contrast to behavior cloning methods that train deterministic policies. By leveraging 3D representation learning, these policies can accurately complete tasks by accounting for the spatial properties of objects, such as orientation and position. This is especially effective for tasks that closely resemble those encountered during training Ze et al. (2023). However, these policies often lack the language understanding and generalization abilities of foundation models. Our method builds upon the diffusion architecture (Ke et al., 2024), enhancing its ability to utilize demonstration data through imitation learning, while integrating foundation models to improve generalization, combining the strengths of both approaches.

**Foundation Models for 3D Manipulation.** Recent foundation models trained on internet-scale data have shown strong zero-shot and few-shot generalization, offering new opportunities for complex 3D manipulation tasks (Hu et al., 2023a; Zhou et al., 2023). While some approaches fine-tune

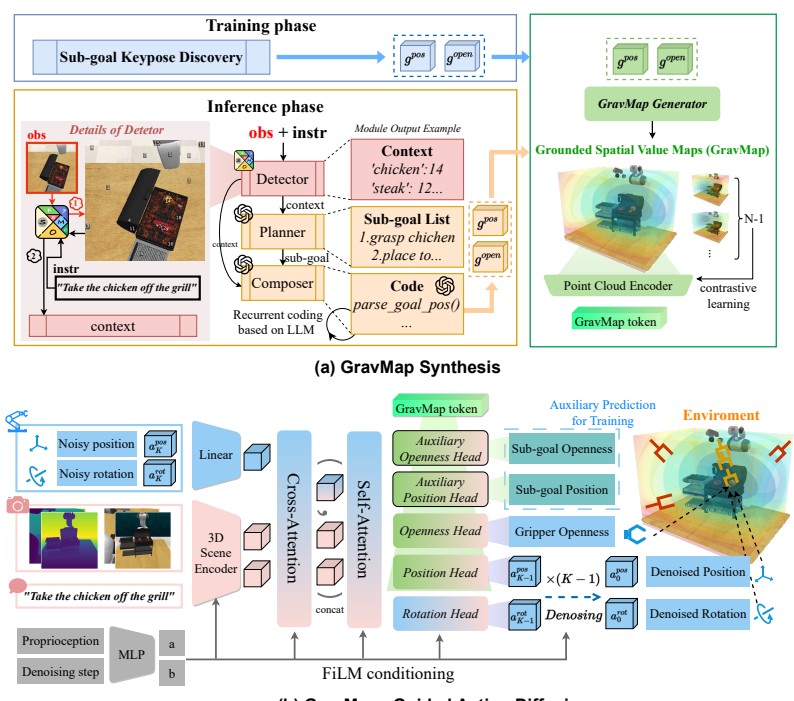

Figure 2: **GravMAD Overview.** (a) **GravMap Synthesis:** During training, we use Sub-goal Keypose Discovery to obtain sub-goals $g^{\text{pos}}$ and $g^{\text{open}}$. During inference, the Detector, Planner, and Composer pipeline interprets visual observations and language instructions to derive $g^{\text{pos}}$ and $g^{\text{open}}$, which are processed into a GravMap and encoded as a GravMap token. (b) **GravMaps Guided Action Diffusion:** The policy network perceives the scene and denoises noisy actions guided by the GravMap token. After $K$ denoising steps, the clean actions are executed by the robot.

vision-language models with embodied data (Driess et al., 2023; Li et al., 2024), this increases computational costs due to the large data requirements. Alternatively, foundational vision models can generate visual representations for 3D manipulation tasks (Zhang et al., 2024; 2023), but they often lack the reasoning capabilities needed for complex tasks. To address these challenges, some studies leverages large language models (LLMs) as high-level planners (Brohan et al., 2023b; Hu et al., 2023b; Huang et al., 2022), generating language-based plans executed by lower-level policies. Others utilize LLMs' code-writing abilities to control robots via API calls or to create value maps for planning robot trajectories (Liang et al., 2023; Huang et al., 2023). However, these methods often sacrifice precision due to a rough understanding of complex 3D scenes. Recent works have combined the reasoning capabilities of foundation models with fine-grained control in 3D manipulation to overcome this limitation (Huang et al., 2024; Sharan et al., 2024). For example, Huang et al. (2024) uses pre-trained vision-language models (VLMs) to provide spatial constraints and a nonlinear solver to generate precise grasp poses. Our method combines the learning power of diffusion architectures with the generalization of VLMs. VLMs generate spatial value maps that guide action diffusion, enabling precise control and multi-task generalization in 3D manipulation tasks.

## 3 METHOD

In this section, we introduce **GravMAD**, a multi-task, sub-goal-driven, language-conditioned diffusion framework for 3D manipulation, as shown in Fig. 2. We divide GravMAD's design into three parts: Section 3.1 defines the problem setting, Section 3.2 explains the definition and generation of GravMaps, and Section 3.3 details how GravMaps guide action diffusion in 3D manipulation.

### 3.1 PROBLEM FORMULATION

We consider a problem setting where expert demonstrations consist of a robot trajectory $(o_1, a_1, o_2, a_2, \ldots)$ and a natural language instruction $\ell \in \mathcal{L}$ that describes the task goal. Each

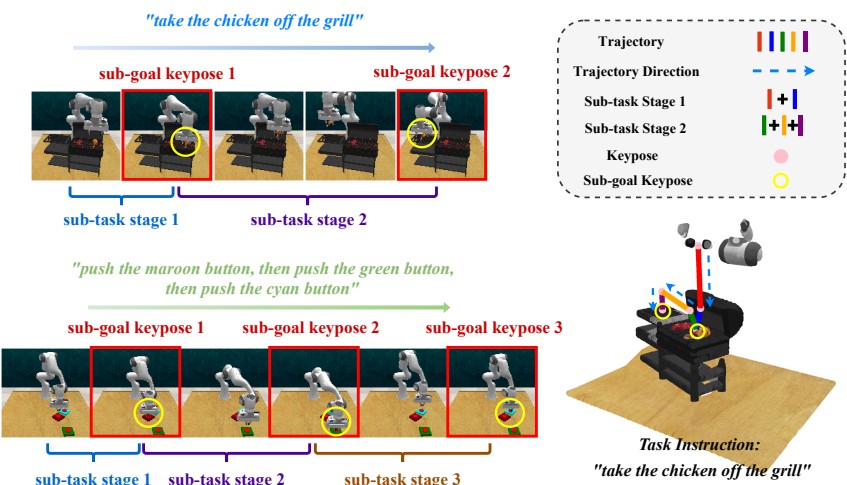

Figure 3: Visualization of sub-goal keyposes and sub-task stages. The left sub-figure shows image-based sub-goal keyposes and sub-task stages for *"take the chicken off the grill"* and *"push the __ button"* tasks. The right shows the sub-goal key poses and sub-task stages in the trajectory for the *"take the chicken off the grill"* task.

observation $o_t \in \mathcal{O}$ includes RGB-D images from one or more viewpoints. Each action $a_t \in \mathcal{A}$ contains the 3D position of the robot's end-effector $a^{pos} \in \mathbb{R}^3$, a 6D rotation $a^{rot} \in \mathbb{R}^6$, and a binary gripper state $a^{open} \in \{0, 1\}$. To address potential discontinuities from quaternion constraints and ensure smooth optimization, we utilize the 6D rotation representation (Ke et al., 2024). In this setting, we assume that a robotic task is composed of multiple sub-tasks, with each sub-task completed when the robot reaches a sub-goal $g_t \in \mathcal{G}$, which specifies the 3D position $g^{pos} \in \mathbb{R}^3$, the gripper openness $g^{open} \in \{0, 1\}$, and the 6D rotation $g^{rot} \in \mathbb{R}^6$. Based on this, we construct a new dataset $\mathcal{D} = \{\zeta_1, \zeta_2, \ldots\}$ from expert demonstrations. Each demonstration $\zeta$ consists of trajectories with sub-goals $\{(o_1, g_1, a_1), (o_2, g_2, a_2), \ldots\}$ and the corresponding language instruction $\ell$. Our goal is to learn a policy $\pi : (\mathcal{O}, \mathcal{L}, \mathcal{G}) \mapsto \mathcal{A}$, which maps observations $o_t$, sub-goals $g_t$, and instructions $\ell$ to actions $a_t$. To facilitate sub-task segmentation and efficiently learn the policy, we frame the robot's 3D manipulation learning problem as a keypose prediction problem following prior works (James & Davison, 2022; James et al., 2022; Goyal et al., 2023; Shridhar et al., 2023). Our model progressively predicts the next keypose based on current observations and uses a sampling-based motion planner (Klemm et al., 2015) to plan the trajectory between two keyposes. In the existing keypose discovery method, a pose is identified as a keypose when the robot's joint velocities are near zero or the gripper state changes (James & Davison, 2022). Our work filters the task's sub-goals based on these keyposes to facilitate sub-task segmentation and ensure efficient completion of the overall task.

## 3.2 GRAVMAP: GROUNDED SPATIAL VALUE MAPS

To tackle generalization challenges in 3D manipulation tasks, we introduce the spatial value maps (**GravMap**), an adaptation of the voxel value maps proposed by Huang et al., denoted as $m$. GravMaps are adaptively synthesized based on task variations, translating language instructions into 3D spatial sub-goals and reflecting the spatial relationships within the environment. This provides precise guidance for robotic action diffusion. Each GravMap $m$ contains two voxel maps: (1) a spatial cost map $m_c$, with lower values near the sub-goal and higher costs further away, and (2) a gripper openness map $m_o$, indicating where the gripper should open or close. As shown in Fig. 2(a), GravMaps are generated differently for training and inference. In training, they are identified from expert demonstrations using the sub-goal keypose discovery method. During inference, pre-trained models generate them from language instructions and observed images.

**GravMap Synthesis with Sub-goal keypose Discovery during Training.** We define each sub-task stage in 3D manipulation as: (1) the process where the robotic end-effector transitions from not touching an object to making contact, or (2) the interaction between the end-effector or tool and a new object, where a series of operations are performed before disengaging. To efficiently segment these

sub-task stages and find sub-goals, we build upon the existing keypose discovery method (James & Davison, 2022) and propose a novel data distillation method called **sub-goal keypose discovery**.

The sub-goal keypose discovery process iterates over each keypose $K_p^i \in \{K_p\}_1^{N_k}$, where $N_k$ is the number of keyposes in a task. For each keypose, the corresponding observation-action pair $(o_{K_p^i}, a_{K_p^i})$ is passed to the function $S_K$, which outputs a Boolean value to determine whether the given keypose should be discovered as a sub-goal keypose. The decision is made based on whether the keypose satisfies the discovery constraint: $S_K((o_{K_p^i}, a_{K_p^i})) = \begin{cases} 1, & \text{if discovery constraints are met} \\ 0, & \text{otherwise} \end{cases}$ . The function $S_K$ can incorporate multiple constraints. In our paper, we define two constraints for $S_K$, depending on the type of manipulation task, as shown in Fig. 3: (1) For grasping tasks, such as *"take the chicken off the grill"*, sub-goal keyposes are discovered based on the following constraints: a change in the gripper's open/close state and a significant change in touch force. (2) For contact-based tasks, such as *"push the __ button"*, sub-goal keyposes are discovered solely based on significant changes in touch force. For more details on sub-goal keypose discovery, please refer to Appendix A.2.

After discovering the sub-goal keyposes, the sub-task stages can be quickly segmented, and the corresponding sub-goals can be identified. The end-effector position $g^{\text{pos}}$ and gripper openness $g^{\text{open}}$ at these sub-goals are then input to the *Gravmap generator* to generate the GravMaps $m$ for training. The process of the GravMap generator is illustrated in Algorithm 1 in Appendix A.1, adapted from Huang et al. (2023).

**GravMap Synthesis with Foundation Model during Inference.** During the inference phase, we use pre-trained foundation models to synthesize GravMaps. First, to enable the robot to tie the task-related words with their manifestation in the 3D environment, we introduce a Set-of-Mark (SoM) (Yang et al., 2023a)-based Detector. This Detector uses Semantic-SAM (Li et al., 2023) to perform semantic segmentation on the observed RGB images and assigns numerical tags to the segmented regions. Next, the Detector uses GPT-4o to select task-relevant objects and their corresponding tags from the labeled images as contextual information $\mathcal{C}$. Based on the task instructions $\ell$ and the context $\mathcal{C}$ provided by the Detector, we apply the LLM-based Planner proposed by Huang et al. to infer a series of text-based sub-goals. Then, an LLM-based Composer (Huang et al., 2023) recursively generates code to parse each sub-goal. During execution, the code uses the context $\mathcal{C}$ to obtain the end-effector positions $g^{\text{pos}}$ and gripper openness states $g^{\text{open}}$ corresponding to each sub-goal. Finally, $g^{\text{pos}}$ and $g^{\text{open}}$ are fed into the *GravMap generator* shown in Algorithm 1, skipping the data augmentation process to generate the GravMaps. Details of this process can be found in Appendix A.3.2.

We synthesize the GravMaps via sub-goal keypose discovery during training or foundation models during inference. GravMaps $m$ are then downsampled using farthest point sampling (FPS) and encoded into token $t_m$ with the DP3 (Ze et al., 2024) encoder, a lightweight MLP network.

### 3.3 GRAVMAPS GUIDED ACTION DIFFUSION

After obtaining the GravMaps, they can be used to guide the action diffusion process, as shown in Fig. 2(b). Before the diffusion process begins, the robot should first perceive the 3D environment.

**3D Scene Perception.** Building on previous works (Gervet et al., 2023; Ke et al., 2024), we use a 3D scene encoder to transform language instructions and multi-view RGB-D images into scene tokens, enhancing the robot's 3D scene perception. RGB images are encoded using a pre-trained CLIP ResNet50 backbone (Radford et al., 2021) and a feature pyramid network. These features are lifted into 3D feature clouds using 3D positions derived from depth images and camera intrinsics. Simultaneously, the CLIP language encoder converts task instructions into language tokens. These tokens interact with the 3D feature cloud to generate scene tokens ($t_s$), enabling the robot to capture 3D environmental information.

**GravMaps Guided Action Diffusion.** GravMAD builds upon the 3D trajectory diffusion architecture introduced by 3D Diffuser Actor (Ke et al., 2024) and further integrates GravMap tokens $t_m$ to guide the action diffusion process. Specifically, GravMAD models policy learning as the reconstruction of the robot's end-effector pose using diffusion probabilistic models (DDPMs) (Ho et al., 2020). The end-effector pose is represented as $e = (a^{\text{pos}}, a^{\text{rot}})$. Starting with Gaussian noise $e_K = (a_K^{\text{pos}}, a_K^{\text{rot}})$, the denoising networks $\epsilon_\theta^{\text{pos}}$ and $\epsilon_\theta^{\text{rot}}$ perform $K$ iterative steps to progressively reconstruct the clean

pose $e_0 = \left(a_0^{\text{pos}}, a_0^{\text{rot}}\right)$:

$$
\begin{aligned}
a_{k-1}^{\text{pos}} &= \alpha \left( a_k^{\text{pos}} - \gamma \epsilon_\theta^{\text{pos}} \left(e_k, k, p, t_s, t_m\right) + \mathcal{N}\left(0, \sigma^2 I\right) \right), \\
a_{k-1}^{\text{rot}} &= \alpha \left( a_k^{\text{rot}} - \gamma \epsilon_\theta^{\text{rot}} \left(e_k, k, p, t_s\right) + \mathcal{N}\left(0, \sigma^2 I\right) \right),
\end{aligned}
\tag{1}
$$

where $\alpha$, $\gamma$, and $\sigma$ are functions of the iteration step $k$, determined by the noise schedule. $\mathcal{N}\left(0, \sigma^2 I\right)$ is Gaussian noise. Here, $p$ represents proprioceptive information (a short action history). The denoising networks use 3D relative position attention layers (Gervet et al., 2023; Xian et al., 2023; Ke et al., 2024), with FiLM (Perez et al., 2018) conditioning applied to each layer based on proprioception $p$ and denoising step $k$. As shown in Fig. 2(b), after passing through linear layers, $a_K^{\text{pos}}$ and $a_K^{\text{rot}}$ are concatenated and attend to the 3D scene tokens $t_s$ via cross-attention. A self-attention layer then refines this representation to produce end-effector contextual features. These features are processed by five prediction heads: the *Position Head*, *Rotation Head*, *Openness Head*, *Auxiliary Openness Head*, and *Auxiliary Position Head*. In all but the rotation head, contextual features undergo cross-attention with GravMap tokens, followed by an MLP to predict the target values. See Appendix A.4 for details.

The first two prediction heads predict the noise added to the original pose using the $L1$ norm, with the losses defined as:

$$
\begin{aligned}
\mathcal{L}_{\text{pos}} &= \| \epsilon_k^{\text{pos}} - \epsilon_\theta^{\text{pos}} \left(e_k, k, p, t_s, t_m\right) \|, \\
\mathcal{L}_{\text{rot}} &= \| \epsilon_k^{\text{rot}} - \epsilon_\theta^{\text{rot}} \left(e_k, k, p, t_s\right) \|,
\end{aligned}
\tag{2}
$$

where iteration $k$ is randomly selected, and $\epsilon_k^{\text{pos}}$ and $\epsilon_k^{\text{rot}}$ are randomly sampled as the ground truth noise.

The third prediction head is used to predict the gripper's open/close state, and we use binary cross-entropy (BCE) loss for supervision:

$$
\mathcal{L}_{\text{open}} = \text{BCE} \left( f_\theta^{\text{open}} \left(e_k, k, p, t_s, t_m\right), a^{\text{open}} \right)
\tag{3}
$$

The last two prediction heads enable GravMAD to better focus on the ideal end-effector pose at sub-goals, with the loss functions defined as follows:

$$
\begin{aligned}
\mathcal{L}_{\text{aux\_pos}} &= \| g^{\text{pos}} - f_\theta^{\text{aux\_pos}} \left(e_k, k, p, t_s, t_m\right) ) \|, \\
\mathcal{L}_{\text{aux\_open}} &= \text{BCE} \left( f_\theta^{\text{aux\_open}}(e_k, k, p, t_s, t_m), g^{\text{open}} \right),
\end{aligned}
\tag{4}
$$

where $f_\theta$ represents the pose prediction network in GravMAD, while $g^{\text{pos}}$ and $g^{\text{open}}$ denote the ground truth sub-goal positions and gripper openness, respectively.

In addition to the losses related to robot actions mentioned above, a contrastive learning loss is applied to enhance feature representations from GravMaps. Positive pairs are features from the same GravMap, while negative pairs come from different GravMaps. In each forward pass, one GravMap is extracted from the dataset, and $N - 1$ different GravMaps are randomly generated. The loss maximizes similarity between positive pairs and minimizes it between negative pairs:

$$
\mathcal{L}_{\text{con}} = -\frac{1}{N} \sum_{i=1}^{N} \log \frac{\exp(f_{g_i} \cdot f_{g_i}^{+} / T)}{\sum_{j=1}^{N} \exp(f_{g_i} \cdot f_{g_j} / T)},
\tag{5}
$$

where $T$ is the temperature parameter, $f_{g_i}$ represents the feature of the $i$-th sample, and $f_{g_i}^{+}$ represents the positive feature of the $i$-th sample.

At this stage, the training objective of GravMAD can be formulated by combining the losses from Eq.2, 3, 4, and 5 as follows:

$$
\mathcal{L}_{\text{GravMAD}} = \mathcal{L}_{\text{open}} + \omega_1 \cdot \mathcal{L}_{\text{pos}} + \omega_2 \cdot \mathcal{L}_{\text{rot}} + \omega_3 \cdot \mathcal{L}_{\text{aux\_pos}} + \mathcal{L}_{\text{aux\_open}} + \omega_4 \cdot \mathcal{L}_{\text{con}},
\tag{6}
$$

where $\omega_1, \omega_2, \omega_3, \omega_4$ are adjustable hyperparameters. For more detailed implementation of GravMap and GravMAD, please refer to Appendix A.

## 4 EXPERIMENTS

We aim to answer the following questions: **(i)** Can GravMAD achieve superior generalization in novel 3D manipulation tasks compared to SOTA models? (See Sec. 4.2) **(ii)** Is GravMAD's performance competitive on the 3D manipulation tasks encountered during training? (See Sec. 4.3) **(iii)** What key design elements contribute significantly to GravMAD's overall performance? (See Sec. 4.4)

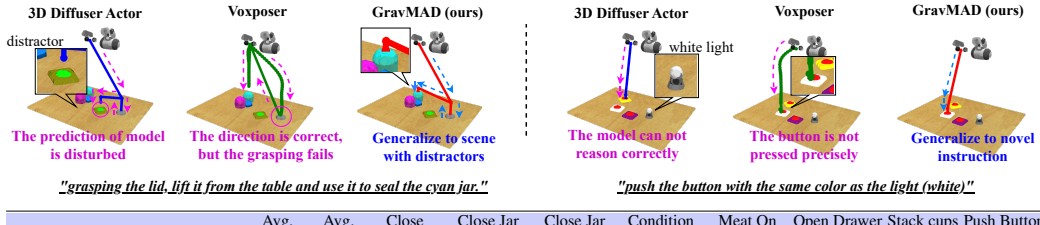

"grasping the lid, lift it from the table and use it to seal the cyan jar."     "push the button with the same color as the light (white)"

| Models | Avg. Success ↑ | Avg. Rank ↓ | Close Drawer | Close Jar Banana | Close Jar Distractor | Condition Block | Meat On Grill | Open Drawer Small | Stack cups blocks | Push Buttons Light |
|---|---|---|---|---|---|---|---|---|---|---|
| Voxposer (Huang et al., 2023) | 34.29 | 2.6 | 96.00 ±4.00 | 17.33 ±19.73 | 22.67 ±10.07 | 25.00 ±23.26 | 38.67 ±12.22 | 6.67 ±2.31 | 0.00 ±0.00 | 68.00 ±18.33 |
| Act3D (Gervet et al., 2023) | 17.83 | 3.5 | 66.67 ±9.24 | 29.33 ±9.24 | 41.33 ±4.62 | 0.00 ±0.00 | 1.33 ±2.31 | 2.67 ±4.62 | 0.00 ±0.00 | 1.33 ±2.31 |
| 3D Diffuser Actor (Ke et al., 2024) | 29.38 | 2.9 | 81.33 ±6.11 | 48.00 ±4.00 | 42.67 ±4.62 | 27.00 ±10.15 | 0.00 ±0.00 | 2.67 ±4.62 | 2.67 ±2.31 | 30.67 ±12.86 |
| GravMAD (VLM) | **62.92** | **1.0** | **97.33** ±2.31 | **84.00** ±00.00 | **86.67** ±2.31 | **74.00** ±11.14 | **45.33** ±4.62 | **21.33** ±12.86 | **18.67** ±2.31 | **76.00** ±8.00 |
| Performance gain | **+28.63** | - | **+1.33** | **+36.00** | **+44.00** | **+47.00** | **+6.66** | **+14.66** | **+16.00** | **+8.00** |

Table 1: **Generalization to 8 novel RLBench tasks.** Evaluations on 8 novel tasks are conducted using 3 seeds, with 25 test episodes per task, utilizing the final checkpoints from training on 12 base tasks. Performance gains are compared to the best-performing baselines, indicated by underlines.

## 4.1 ENVIRONMENTAL SETUP

To thoroughly investigate these questions, we conduct our experiments on a representative instruction-following 3D manipulation benchmark, RLBench (James et al., 2020). Simulation experiments are conducted on two types of tasks to provide a comprehensive evaluation of GravMAD. **1) Base tasks.** To evaluate GravMAD's performance across 3D manipulation tasks encountered during training, we select 12 base tasks from RLBench's 100 language-conditioned tasks, each featuring 2 to 60 variations in instructions, such as handling objects of different colors or quantities. For each base task, we collect 20 demonstrations for training and evaluate the final checkpoints using 3 random seeds over 25 episodes. Detailed descriptions of these tasks are provided in Appendix B.1. **2) Novel tasks.** To further test GravMAD's generalization capabilities, we modify the scene configurations or task instructions of several base tasks to create 8 novel tasks across 3 novelty categories as illustrated in fig. 10. These modifications introduce significant challenges for the robot regarding instruction comprehension, environmental perception, and policy generalization, as described in Appendix B.2. For each novel task, we evaluate the final checkpoints trained on the 12 base tasks. We use 3 random seeds over 25 episodes for each novel task. For all tasks, we use a front-view $256 \times 256$ RGB-D camera and a Franka Panda robot with parallel grippers. Additionally, we further validate GravMAD on 10 real-world robotic tasks, with details provided in Appendix D.6.

**Baselines.** We compare GravMAD against various baselines, covering both foundation model-based and imitation learning-based methods. For the foundation model-based approach, we use **VoxPoser** (Huang et al., 2023) as the baseline. VoxPoser leverages GPT-4 to generate code for constructing value maps, which are then used by a heuristic-based motion planner to synthesize robotic arm trajectories. We reproduce this baseline in our tasks using prompt templates from Huang et al. and our SoM-based Detector, with five camera viewpoints in RLBench. For the imitation learning-based baselines, we select: (1) **3D Diffuser Actor** (Ke et al., 2024), which combines 3D scene representations with a diffusion policy for robotic manipulation tasks. To highlight instruction-following tasks, we use the enhanced language-conditioned version provided by Ke et al.; and (2) **Act3D** (Gervet et al., 2023), which uses a 3D feature field within a policy transformer to represent the robot's workspace. Differences between GravMAD and these baselines are detailed in Appendix A.5.

**Training and Evaluation Details.** GravMAD runs in a multi-task setting during both the training and testing phases. All models complete 600k training iterations on an NVIDIA RTX4090 GPU, with the final checkpoint selected using three random seeds for evaluation. During testing, except for the novel task *"push buttons light"*, which must be completed in 3 time steps, all other tasks must be completed in 25 time steps; otherwise, they are considered failures. Evaluation metrics include the average success rate and rank. The success rate measures the proportion of tasks completed according to language instructions. Meanwhile, the average rank calculates the average of the rankings of each model in all tasks, reflecting the overall performance of the model in the tasks. Two settings are used to generate context $\mathcal{C}$ during testing: *Manual* and *VLM*. In the manual setting, we manually provide the Detector with the precise 3D coordinates of task-related objects in the simulation to generate accurate context. In the VLM setting, we use a Detector implemented with SoM and GPT-4o to locate task-related objects and generate context.

| Models | Avg. Success ↑ | Avg. Rank ↓ | Close Jar | Open Drawer | Meat off Grill | Slide Block | Put in Drawer |
|---|---|---|---|---|---|---|---|
| Voxposer (Huang et al., 2023) | 15.11 | 4.5 | $12.00_{\pm 10.58}$ | $10.67_{\pm 8.33}$ | $45.33_{\pm 24.44}$ | $0.00_{\pm 0.00}$ | $0.00_{\pm 0.00}$ |
| Act3D (Gervet et al., 2023) | 34.11 | 4.3 | $61.33_{\pm 4.62}$ | $41.33_{\pm 4.62}$ | $60.0_{\pm 6.92}$ | $78.67_{\pm 2.31}$ | $49.33_{\pm 10.07}$ |
| 3D Diffuser Actor (Ke et al., 2024) | 55.81 | 2.3 | $66.67_{\pm 2.31}$ | $88.00_{\pm 6.93}$ | $88.00_{\pm 4.00}$ | $84.00_{\pm 0.00}$ | $94.67_{\pm 2.31}$ |
| GravMAD (Manual) | **69.17** | 1.3 | **$100.00_{\pm 0.00}$** | $76.67_{\pm 4.62}$ | $89.33_{\pm 2.31}$ | $93.33_{\pm 2.31}$ | $78.67_{\pm 6.11}$ |
| Performance gain | +13.36 | - | +33.33 | -13.33 | +1.33 | +9.33 | -16.00 |
| GravMAD (VLM) | **56.72** | 2.1 | **$100.00_{\pm 0.00}$** | $58.67_{\pm 2.31}$ | $70.67_{\pm 2.31}$ | $80.00_{\pm 0.00}$ | $61.33_{\pm 9.24}$ |
| Performance gain | +0.91 | - | +33.33 | -29.33 | -17.33 | -4.00 | -33.34 |
| Models | Push Buttons | Stack Blocks | Place Cups | Place Wine | Screw Bulb | Insert Peg | Stack Cups |
| Voxposer (Huang et al., 2023) | $80.00_{\pm 13.86}$ | $16.00_{\pm 12.00}$ | $6.67_{\pm 8.33}$ | $5.33_{\pm 2.31}$ | $4.00_{\pm 4.00}$ | $0.00_{\pm 0.00}$ | $1.33_{\pm 2.31}$ |
| Act3D (Gervet et al., 2023) | $66.67_{\pm 2.31}$ | $0.00_{\pm 0.00}$ | $0.00_{\pm 0.00}$ | $45.33_{\pm 2.31}$ | $6.67_{\pm 2.31}$ | $0.00_{\pm 0.00}$ | $0.00_{\pm 0.00}$ |
| 3D Diffuser Actor (Ke et al., 2024) | $94.67_{\pm 2.31}$ | $13.67_{\pm 2.89}$ | $5.33_{\pm 6.11}$ | $82.67_{\pm 2.31}$ | $29.33_{\pm 2.31}$ | $2.67_{\pm 4.62}$ | $20.00_{\pm 0.00}$ |
| GravMAD (Manual) | **$98.67_{\pm 2.31}$** | **$56.67_{\pm 4.62}$** | $5.33_{\pm 2.31}$ | $77.33_{\pm 4.62}$ | $66.67_{\pm 6.11}$ | $32.00_{\pm 6.93}$ | $57.33_{\pm 2.31}$ |
| Performance gain | +4.00 | +40.67 | -1.34 | -5.34 | +37.34 | +29.33 | +37.33 |
| GravMAD (VLM) | **$97.33_{\pm 2.31}$** | **$51.33_{\pm 6.11}$** | $5.33_{\pm 4.62}$ | $33.33_{\pm 4.62}$ | $54.67_{\pm 6.11}$ | $18.67_{\pm 4.62}$ | $49.33_{\pm 2.31}$ |
| Performance gain | +2.66 | +35.33 | -1.34 | -49.34 | +25.34 | +16.00 | +29.33 |

Table 2: **Multi-task test results on 12 base tasks.** All models are trained on 12 base tasks with 20 demonstrations each. Final checkpoints are evaluated across 3 seeds with 25 test episodes per task. Performance gains are compared to the best-performing baselines.

## 4.2 GENERALIZATION PERFORMANCE OF GRAVMAD TO NOVEL TASKS

In Table 1, we present the generalization performance of models trained on 12 base tasks when tested on 8 novel tasks, along with visualized trajectories from two of these tasks. The results show that changes in task scenarios and instructions negatively impact the test performance of all pre-trained models to some extent. However, GravMAD exhibits superior generalization across all 8 novel tasks compared to the baseline models. In terms of average success rate, GravMAD outperforms VoxPoser, Act3D, and 3D Diffuser Actor by **28.63%**, **45.09%**, and **33.54%**, respectively. VoxPoser leverages large models to achieve a certain level of performance on novel tasks, but its heuristic motion planner fails to grasp object properties and task interaction conditions, leading to poor results on tasks requiring fine manipulation, as shown in the trajectory visualizations. Similarly, 3D Diffuser Actor and Act3D struggle to transfer skills from training to novel tasks, primarily due to overfitting to training-specific tasks, which hampers generalization. In contrast, GravMAD uses VLM-generated GravMaps to guide action diffusion, enabling effective object interaction and strong performance on novel tasks. These results clearly demonstrate GravMAD's superior generalization.

## 4.3 TEST PERFORMANCE OF GRAVMAD ON BASE TASKS

Table 2 compares the performance of all models on 12 base tasks. GravMAD (Manual) outperforms Act3D and Voxposer across all tasks and exceeds the best baseline, 3D Diffuser Actor, in 9 out of 12 tasks, with an average success rate improvement of **13.36%**. Despite the Detector's coarse SoM positioning affecting GravMAD (VLM)'s performance, it still outperforms Act3D and Voxposer on all tasks, with a **0.91%** higher average success rate than 3D Diffuser Actor. These results clearly show that GravMAD remains highly competitive even on previously seen tasks. As long as task-related object positions are accurate, the generated GravMap effectively reflects sub-goals and guides action diffusion, enabling precise execution by GravMAD. GravMAD (Manual) underperforms 3D Diffuser Actor in the *"open drawer"*, *"put in drawer"*, and *"place wine"* tasks due to slight deviations between the manually provided object positions and the sub-goals. In high-precision tasks, even small deviations can impact performance. For example, in the *"open drawer"* task, the robot needs to grasp the center of the small handle for optimal performance. After manually adjusting the sub-goal to better align with the handle, performance improved. GravMAD (VLM) also struggles in tasks like *"Place Wine"* due to inaccuracies in the object positions provided by the Detector, especially when Semantic SAM fails to provide precise locations or the camera doesn't capture the full scene. For further analysis of failure cases, please refer to Appendix B.3.

## 4.4 ABLATIONS

Extensive ablation studies are conducted to analyze the role of each key design element in GravMAD, with the results shown in Fig 4. The following findings are revealed: **1) Impact of replacing GravMaps with specific sub-goal position and openness**: Replacing GravMaps with sub-goals $g^{\text{pos}}$ and $g^{\text{open}}$ (w/o GravMap) results in a significant performance drop. Without GravMaps, the policy lacks regional context, becoming overly sensitive to precise positions and unable to generalize to

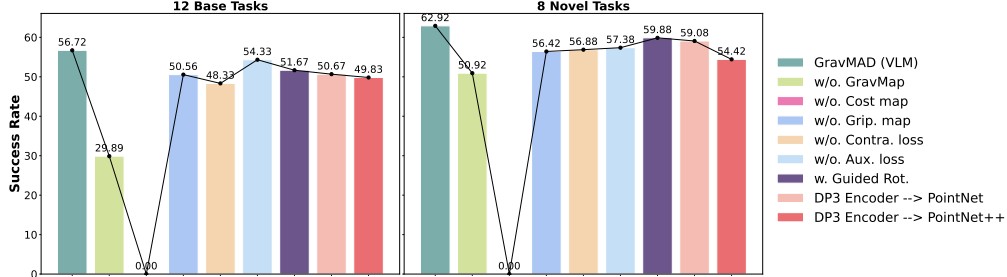

Figure 4: **Ablation Studies**. We evaluate the impact of key design elements by reporting the average success rates across 12 base tasks and 8 novel tasks. In the results, "→" denotes replacement, "w/o" indicates "without", and "w." signifies "with".

slight spatial variations. **2) Importance of both cost map and gripper map in GravMaps**: The combination of the cost map and gripper map within GravMaps is essential for guiding the model's attention to sub-goal locations and ensuring effective gripper usage. The absence of the gripper map causes a moderate decline in performance (w/o. Grip. map). In contrast, omitting the cost map causes zero-gradient issues during training, leading to incorrect predictions and task failure. This occurs because the encoder cannot process such input. Additional experiments for this ablation, detailed in Appendix D.5, highlight the cost map's impact on performance. (w/o. Cost map) **3) Significance of contrastive learning loss and auxiliary losses**: Removing the contrastive learning loss $\mathcal{L}_{con}$ results in highly similar features from the point cloud encoder, diminishing their effectiveness in action denoising and leading to a decline in model performance (w/o. Contra. loss). Similarly, the absence of auxiliary losses $\mathcal{L}_{aux\_pos}$ and $\mathcal{L}_{aux\_open}$ weakens the model's focus on sub-goals, leading to a noticeable drop in performance (w/o Aux. loss). **4) Effect of GravMap tokens on guiding rotation actions:** Conditioning rotation actions with GravMap tokens in the action diffusion process results in a performance drop, likely due to the inherent nature of rotation actions, which makes them difficult to be guided explicitly through value maps (w. Guided Rot.). **5) Impact of different point cloud encoders on GravMap performance.** Replacing the DP3 encoder in GravMAD with PointNet (Qi et al., 2017a) (DP3 Encoder → PointNet) or PointNet++ (Qi et al., 2017b) (DP3 Encoder → PointNet++) leads to a performance decline. We suspect that lightweight encoders help prevent overfitting to training data details, enhancing GravMAD's generalization ability across different tasks or unseen data.

## 5 CONCLUSION AND DISCUSSION

In this paper, we introduce GravMAD, a novel action diffusion framework that facilitates generalized 3D manipulation using sub-goals. GravMAD grounds language instructions into spatial subgoals within the 3D workspace through grounded spatial value maps (GravMaps). During training, these GravMaps are generated from demonstrations by Sub-goal Keyposes Discovery. In the inference phase, GravMaps are constructed by leveraging foundational models to directly predict sub-goals. Consequently, GravMAD seamlessly integrates the precision of imitation learning with the strong generalization capabilities of foundational models, leading to superior performance across a variety of manipulation tasks. Extensive experiments on the RLBench benchmark and real-robot tasks show that GravMAD achieves competitive performance on training tasks. It also generalizes well to novel tasks, demonstrating its potential for practical use across diverse 3D environments. Despite its promising results, GravMAD has some limitations. First, its effectiveness is highly dependent on prompt engineering, which can be challenging for inexperienced users. Additionally, visual-language models (VLMs) have limited detection capabilities and are sensitive to changes in camera perspective, affecting performance, and preventing optimal efficiency and accuracy. Future work will address these issues to enhance the performance of the model, expand its applicability, and validate its use on more complex and long-horizon real-robot tasks.

## 6 ACKNOWLEDGMENTS

This work was supported in part by the National Natural Science Foundation of China under Grant 62276128, Grant 62192783, Grant 62206166; in part by the Jiangsu Science and Technology Major Project BG2024031; in part by the Natural Science Foundation of Jiangsu Province under Grant BK20243051; in part by the Shanghai Sailing Program under Grant No.23YF1413000; in part by the Postgraduate Research & Practice Innovation Program of Jiangsu Province under Grant KYCX24_0263; in part by the Fundamental Research Funds for the Central Universities (14380128); in part by the Collaborative Innovation Center of Novel Software Technology and Industrialization.

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

# A  ADDITIONAL IMPLEMENTATION DETAILS

## A.1  GRAVMAP GENERATION PROCESS

---

**Algorithm 1:** GravMap Generation Process

---

**Input:** End-effector position $g^{\text{pos}}$, initial gripper openness $g_{\text{init}}^{\text{open}}$, target gripper openness $g^{\text{open}}$, map size $S_m$, offset range $\beta_o$, radius $\beta_r$, downsample ratio $\beta_d$, number of sampled points $N_p$, inference mode flag $inference$

**Output:** GravMap $m$

**begin**

   // Initialize cost map $m_c$, gripper map $m_g$, and avoidance map $m_a$ with size $S_m^3$
   Initialize $m_c$, $m_g$, and $m_a$ with shape $S_m \times S_m \times S_m$, setting $m_c(u,v,w) = 1$,
   $m_g(u,v,w) = g_{\text{init}}^{\text{open}}$, and $m_a(u,v,w) = 0$ for all voxels $(u,v,w)$;

   // Convert $g^{\text{pos}}$ from world coordinates $(x,y,z)$ to voxel coordinates $(i,j,k)$
   Extract $(x,y,z) \leftarrow g^{\text{pos}}$; Convert $(x,y,z)$ to voxel coordinates $(i,j,k)$;

   // Determine voxel coordinates $(i',j',k')$ based on mode
   **if** *not inference* **then**

      // Apply random offsets $\delta_i, \delta_j, \delta_k$ to $(i,j,k)$ for data augmentation
      Sample $\delta_i, \delta_j, \delta_k \sim \text{Uniform}(-\beta_o, \beta_o)$;
      Update voxel coordinates: $(i',j',k') = (i + \delta_i, j + \delta_j, k + \delta_k)$;

   **else**

      // Use original voxel coordinates in inference mode
      $(i',j',k') = (i,j,k)$;

   // Compute Euclidean distance from $(i',j',k')$ for all $(u,v,w)$
   For each voxel $(u,v,w)$, compute $D(u,v,w) = \sqrt{(u-i')^2 + (v-j')^2 + (w-k')^2}$

   // Construct avoidance map $m_a$
   Set $m_a(u,v,w) = 1$ for all occupied voxels in the scene;

   // Update $m_a$ by excluding voxels near the target $(i',j',k')$
   Set $m_a(u,v,w) = 0$ for voxels where $D(u,v,w) < 0.15 \cdot S_m$;
   Smooth $m_a$ with Gaussian filter ($\sigma = 10$);

   // Compute and normalize the cost map $m_c$
   Set $m_c(u,v,w) = \frac{D(u,v,w)}{\max D}$ for all voxels $(u,v,w)$;

   // Combine $m_c$ and $m_a$ into the final cost map
   Update $m_c(u,v,w) = 2 \cdot m_c(u,v,w) + m_a(u,v,w)$ for all voxels $(u,v,w)$;
   Normalize $m_c$ to the range $[0,1]$;

   // Set $m_g$ within radius $\beta_r$ of $(i',j',k')$
   Set $m_g(u,v,w) = g^{\text{open}}$ for voxels where $D(u,v,w) \leq \beta_r$;

   // Downsample both $m_c$ and $m_g$
   Downsample $m_c$ and $m_g$ by $\beta_d$;
   Select $N_p$ points $\{v_p\}$ from the downsampled $m_c$ and $m_g$ using Farthest Point Sampling;

   // Construct GravMap $m$ using sampled points
   Form $m = \{(v_p, m_c(v_p), m_g(v_p))\}_{p=1}^{N_p}$;
   **return** $m$;

---

## A.2  HEURISTICS FOR SUB-GOAL KEYPOSE DISCOVERY

Building on keypose discovery (James & Davison, 2022), we propose the Sub-goal Keypose Discovery method to identify sub-goal keyposes from demonstrations, focusing on changes in the gripper's state and touch forces. This is particularly relevant for object manipulation tasks, where the robot's interactions with objects can be segmented into discrete sub-goals.

The implementation of the Sub-goal Keypose Discovery algorithm starts with a set of pre-computed keyposes, which are frames selected from the demonstration sequence through an initial keypose discovery process. We introduce two functions: `touch_change`, shown in Algorithm 2, and `gripper_change`, shown in Algorithm 3, to evaluate whether a keypose qualifies as a sub-goal.

The first function checks for significant changes in the gripper's touch forces, while the second evaluates changes in the gripper's open/close state. The pseudocode in Algorithm 4 outlines the heuristic steps for identifying sub-goal keyposes.

One current limitation of the Sub-goal Keypose Discovery method is its inability to effectively handle tasks involving tool use, which we plan to address in future research.

---

**Algorithm 2:** touch_change Function

---

**Input:** Demonstration sequence *demo*, Keypose index *k*, Threshold *touch_threshold*, Tolerance *delta*
**Output:** Boolean indicating significant touch force change
**begin**
    Set *start* to max(0, *k - touch_threshold*);
    **for** *each index j from start to k-1* **do**
        **if** *Touch forces at j differ from Touch forces at k within tolerance delta* **then**
            **return** *True*;

    **return** *False*;

---

**Algorithm 3:** gripper_change Function

---

**Input:** Demonstration sequence *demo*, Keypose index *k*, Threshold *gripper_threshold*
**Output:** Boolean indicating gripper state change
**begin**
    Set *start* to max(0, *k - gripper_threshold*);
    **for** *each index j from start to k-1* **do**
        **if** *Gripper state at j differs from Gripper state at k* **then**
            **return** *True*;

    **return** *False*;

---

**Algorithm 4:** Heuristics for Sub-goal Keypose Discovery

---

**Input:** Demonstration sequence *demo*, Task type *task_str*, Threshold parameters *touch_threshold, gripper_threshold, delta*
**Output:** List of sub-goal keyposes *sub_goal_keyposes*
**begin**
    Initialize *sub_goal_keyposes* as an empty list;
    Identify keyposes from *demo* using keypose discovery method;
    **for** *each keypose k in keyposes* **do**
        **if** *task_str is a task involving touch without grasping* **then**
            **if** *touch_change(demo, k, touch_threshold, delta)* **then**
                Append *k* to *sub_goal_keyposes*;
        **else**
            **if** *gripper_change(demo, k, gripper_threshold)* **or** *touch_change(demo, k, touch_threshold, delta)* **then**
                Append *k* to *sub_goal_keyposes*;

    Append the last keypose to *sub_goal_keyposes*;
    **return** *sub_goal_keyposes*;

---

**Algorithm 5:** GravMap Generation

---

**Input:** Instruction $\ell$, Observed RGB Image $\mathcal{O}$

**Prompt :** Prompt for Detector $\mathcal{P}_{\text{det}}$, Prompt for Planner $\mathcal{P}_{\text{plan}}$, Prompt for Composer $\mathcal{P}_{\text{com}}$, Few-shot task specified prompt $\mathcal{P}_{\text{task}} = \{\mathcal{P}'_{\text{det}}, \mathcal{P}'_{\text{plan}}, \mathcal{P}'_{\text{com}}\}$, Cost Map Prompt $\mathcal{P}_{\text{cost}}$, Gripper Map Prompt $\mathcal{P}_{\text{gripper}}$

**Output:** GravMap $m$

**begin**

    $\mathcal{O}' \leftarrow$ Semantic-SAM($\mathcal{O}$); // Label objects with numerical tags

    $\mathcal{C} \leftarrow$ Detector($\ell, \mathcal{O}', \mathcal{P}_{\text{det}}, \mathcal{P}'_{\text{det}}$); // Select relevant objects and get corresponding 3D positions as context

    $ST \leftarrow$ Planner($\ell, \mathcal{C}, \mathcal{P}_{\text{plan}}, \mathcal{P}'_{\text{plan}}$); // Infer sub-tasks $ST = (st_1, st_2, \ldots, st_i)$

    Function calls, parameters $\leftarrow$ Composer($ST, \mathcal{C}, \mathcal{P}_{\text{com}}, \mathcal{P}'_{\text{com}}$); // Generate API calls and their parameters for generating $g^{\text{pos}}$ and $g^{\text{open}}$

    $g^{\text{pos}} \leftarrow$ get_cost_map(Function calls, parameters, $\mathcal{P}_{\text{cost}}$);

    $g^{\text{open}} \leftarrow$ get_gripper_map(Function calls, parameters, $\mathcal{P}_{\text{gripper}}$);

    $m \leftarrow$ GravMap generator(cat($g^{\text{pos}}, g^{\text{open}}$));

    **return** $m$;

---

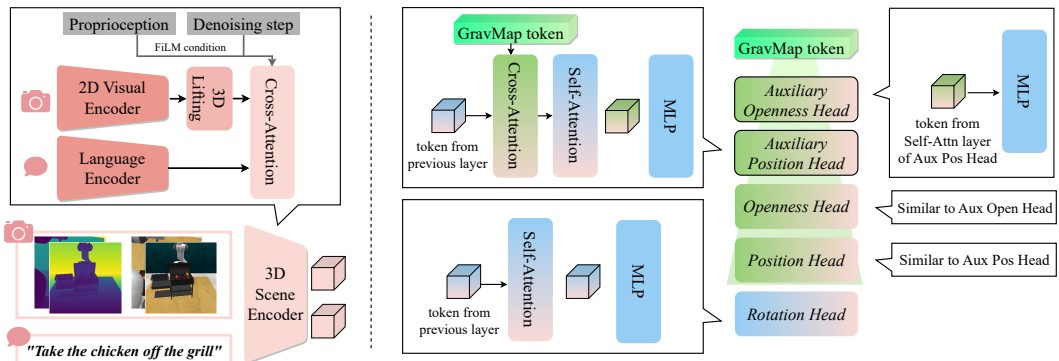

Figure 5: **Detailed description of the modules in GravMAD**, including the 3D Scene Encoder and the prediction heads

## A.3 DETAILS OF GRAVMAP SYNTHESIS

### A.3.1 TRAINING PHASE

To facilitate GravMap synthesis, we assign a goal action to each keypose by linking it to the action performed at the nearest future sub-goal. This association enables us to determine the relevant cost and gripper state for different regions of the GravMap. In the first map, $m_c \in \mathbb{R}^{w \times h \times d}$, the cost is lower near the positions of the robotic end-effector at these sub-goal keyposes and higher as the distance increases. In the second map, $m_o \in \mathbb{R}^{w \times h \times d}$, areas near the end-effector's position at the sub-goal keyposes reflect the gripper state at the sub-goal, while other areas reflect the gripper state at the current frame.

### A.3.2 INFERENCE PHASE

In this section, we introduce the complete pipeline for GravMap generation, as outlined in Algorithm 5. This includes Algorithm 1, which details the process of generating a GravMap from a language instruction $\ell$ and an observed RGB image $\mathcal{O}$. The GravMap generation pipeline integrates VLMs to interpret instructions, ground them in the visual context, and translate them into coarse 3D voxel representations, i.e., the GravMap.

Our pipeline consists of the following three components:

- **Detector.** Starting with an instruction $\ell$ and an observed RGB image $\mathcal{O}$, the RGB image is passed through the Semantic-SAM segmentation model, which labels each object with a numerical tag, producing a labeled image $\mathcal{O}'$. The GPT4o-based Detector uses the prompts $\mathcal{P}_{\text{det}}$ and $\mathcal{P}'_{\text{det}}$ (adapted from Huang et al. (2024)) to select relevant objects and obtain their 3D positions. The output is a set of selected objects, or context $\mathcal{C}$, which includes the objects' identities and their spatial coordinates in the 3D environment. In the VLM setting, the Detector accesses initial RGB images from four views: wrist, left shoulder, right shoulder, and front camera. In the manual setting, precise 3D object attributes are provided from the simulation.

- **Planner.** The GPT4o-based Planner takes the instruction $\ell$, context $\mathcal{C}$, and planner-specific prompts $\mathcal{P}_{\text{plan}}$ and $\mathcal{P}'_{\text{plan}}$ (adapted from Huang et al. (2023)) to infer a sequence of sub-tasks $(st_1, st_2, \ldots, st_i)$. Each sub-task describes an action or interaction needed to fulfill the instruction $\ell$. Progress is tracked based on the robot's gripper state (open/closed) and whether it is holding an object. The current sub-task is then passed to the Composer for further processing.

- **Composer.** Following Huang et al. (2023), the GPT4o-based Composer parses each inferred sub-task $st_i$ using corresponding prompts $\mathcal{P}_{\text{com}}$ and $\mathcal{P}'_{\text{com}}$. The Composer generates the sub-goal position $g^{\text{pos}}$ and sub-goal openness $g^{\text{open}}$ by recursively generating code. This includes calls to `get_cost_map` and `get_gripper_map`, which are triggered by cost map prompt $\mathcal{P}_{\text{cost}}$ and gripper map prompt $\mathcal{P}_{\text{gripper}}$. For example, for a sub-task like "push close the topmost drawer," the Composer might generate: `get_cost_map('a point 30cm into the topmost drawer handle')` and `get_gripper_map('close everywhere')`. Natural language parameters are parsed by GPT to generate code that assigns values to $g^{\text{pos}}$ and $g^{\text{open}}$. The final GravMap generator in Algorithm 1 then processes $g^{\text{pos}}$ and $g^{\text{open}}$ to generate the GravMap $m$.

The prompts mentioned above can be found on the website: `https://gravmad.github.io`.

## A.4 DETAIL OF MODEL ARCHITECTURE AND HYPER-PARAMETERS FOR GRAVMAD

The detailed hyperparameters of GravMAD are listed in Table 3. Additionally, Fig. 5 provides a detailed overview of GravMAD's modules, including the 3D Scene Encoder and the prediction heads.

**(a)** The 3D Scene Encoder processes visual and language information separately, merging them via a cross-attention mechanism, with proprioception integrated through FiLM. This allows the model to understand tasks like *"Take the chicken off the grill"* in a 3D environment. First, the visual input is processed by a 2D Visual Encoder, transforming image data into feature representations. These 2D features are then passed through a 3D lifting module, converting them into 3D representations using depth information. Simultaneously, the language input, such as the instruction *"Take the chicken off the grill"*, is encoded into language tokens by the Language Encoder. Finally, the 3D visual features and language tokens are combined through cross-attention, producing 3D Scene tokens.

**(b)** Each prediction head consists of Attention layers and an MLP. The Auxiliary Position Head receives tokens from the previous layer, which first go through cross-attention with GravMap tokens, followed by self-attention to refine the features. The tokens are then passed through an MLP to output the sub-goal end-effector position. Similarly, the Auxiliary Openness Head takes tokens from the self-attention layer of the Auxiliary Position Head and uses an MLP to predict the sub-goal gripper openness. The Position Head follows the same process as the Auxiliary Position Head, while the Openness Head mirrors the Auxiliary Openness Head. The Rotation Head processes tokens with self-attention and an MLP to predict rotation error.

## A.5 COMPARISON BETWEEN GRAVMAD AND OTHER BASELINE MODELS

We compare GravMAD with Voxposer (Huang et al., 2023) and 3D Diffuser Actor (Ke et al., 2024) in Fig. 6.

**(a) Voxposer.** We describe our reproduction of Voxposer on RLBench. Voxposer uses our SOM-driven Detector to process the input observation and instruction, generating context information. The

|  | Values |
|---|---|
| **Sub-goal Keypose Discovery** |  |
| touch_threshold | 2 |
| Tolerance: delta | 0.005 |
| gripper_threshold | 4 |
| **GravMap** |  |
| map_size: $S_m$ | 100 |
| offset_range: $\beta_o$ | 3 |
| radius: $\beta_r$ | 3 |
| downsample ratio: $\beta_d$ | 4 |
| number of sampled points: $N_p$ | 1024 |
| **Model** |  |
| image_size | 256 |
| token_dim | 120 |
| diffusion_timestep | 100 |
| noise_scheduler: position | scaled_linear |
| noise_scheduler: rotation | squaredcos |
| action_space | absolute pose |
| **Train** |  |
| batch_size | 8 |
| optimizer | Adam |
| train_iters | 600K |
| learning_rate | $1e^{-4}$ |
| weight_decay | $5e^{-4}$ |
| loss weight: $\omega_1$ | 30 |
| loss weight: $\omega_2$ | 10 |
| loss weight: $\omega_3$ | 30 |
| loss weight: $\omega_4$ | 10 |
| **Evaluation** |  |
| maximal step except push_button_light | 25 |
| maximal step of push_button_light | 3 |

Table 3: **Hyper-parameters for GravMAD**, including Sub-goal Keypose Discovery, GravMap, model configuration, training, and evaluation.

Planner then receives this context and outputs a sub-goal, representing an intermediate step necessary for the overall motion plan. The Composer processes this sub-goal, producing three maps: Cost Map, Rotation Map, and Gripper Map. These maps guide the robot's movement toward the target in the environment. Note that Voxposer's testing process involves a different number of steps compared to 3D Diffuser Actor and GravMAD, completing only after all LLM inferences are executed.

**(b) 3D Diffuser Actor.** We use the language-enhanced version of 3D Diffuser Actor as a baseline. 3D Diffuser employs a 3D Scene Encoder to transform visual and language inputs into 3D Scene tokens, providing an understanding of the 3D environment. An MLP encodes noisy estimates of position and rotation into corresponding tokens, which are then fed, along with the 3D Scene tokens, into a denoising network for action diffusion. This network, conditioned on proprioception and the denoising step, includes attention layers, Openness Head, Position Head, and Rotation Head. During diffusion, noisy position/rotation tokens attend to 3D Scene tokens, and cross-attention with instruction tokens enhances language understanding. These instruction tokens are also used in the prediction processes of the Openness, Position, and Rotation heads.

**(c) GravMAD (ours).** GravMAD shares components with Voxposer, such as the Detector, Planner, and Composer, but incorporates task-specific prompt engineering. Unlike Voxposer, which uses maps for planning, GravMAD encodes these maps into tokens using a point cloud encoder, which are then employed in the action diffusion process. Compared to 3D Diffuser Actor, the key difference is that GravMAD uses GravMap tokens instead of language tokens, improving generalization. Additionally, GravMAD introduces two auxiliary tasks to predict sub-goals, enhancing representation learning.

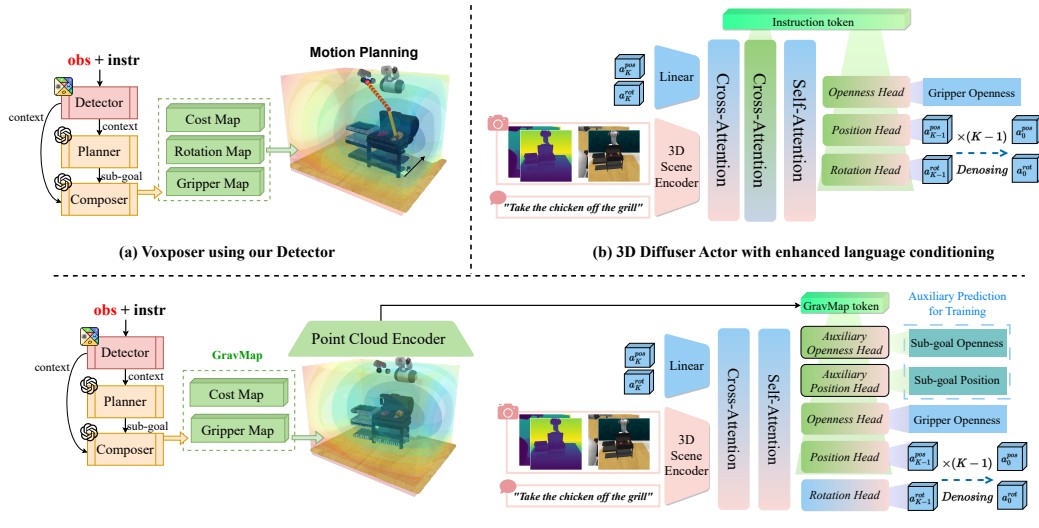

Figure 6: **Comparison of GravMAD with Voxposer and 3D Diffuser Actor**. Unlike Voxposer, which uses planning, GravMAD leverages GravMaps for learning. Compared to 3D Diffuser Actor, GravMAD employs GravMap tokens to guide the action diffusion process and introduces auxiliary position and openness heads to improve representation learning.

# B ADDITIONAL EXPERIMENTAL DETAILS

## B.1 BASE TASK

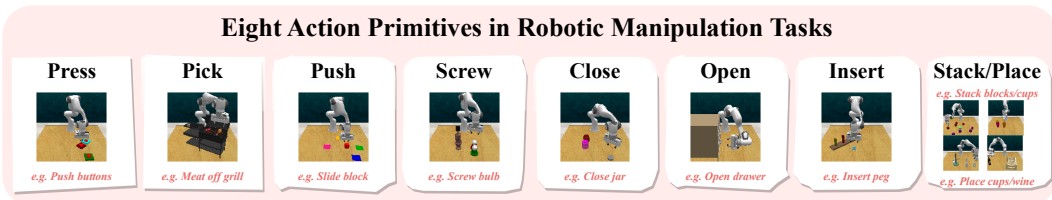

Figure 7: Eight action primitives in robotic manipulation tasks.

For the selection of base tasks, our primary criterion is to ensure they comprehensively cover the fundamental action primitives in robotic manipulation tasks. Therefore, we follow Garcia et al. (2024) and further summarize the eight essential action primitives required for robotic manipulation, as shown in Fig. 7. In line with this criterion, we select 12 base tasks from RLBench (James et al., 2020), as illustrated in Fig. 8. These 12 tasks also include short-term tasks (*close jar, open drawer, meat off grill, slide block, push buttons, place wine*), long-horizon tasks (*put item in drawer, stack blocks, stack cups*), and tasks that require high-precision manipulation (*screw bulb, insert peg, place cups*). Each base task contains 2 to 60 variants in the instructions, covering differences in color, placement, category, and count. In addition to instruction variations, the objects, distractors, and their positions and scenes are randomly initialized in the environment. The templates representing task goals in the instructions are also modified while maintaining their semantic meaning. A summary of the 12 tasks is provided in Table 4.

We provide a detailed description of each task below and explain modifications from RLBench origin codebase.

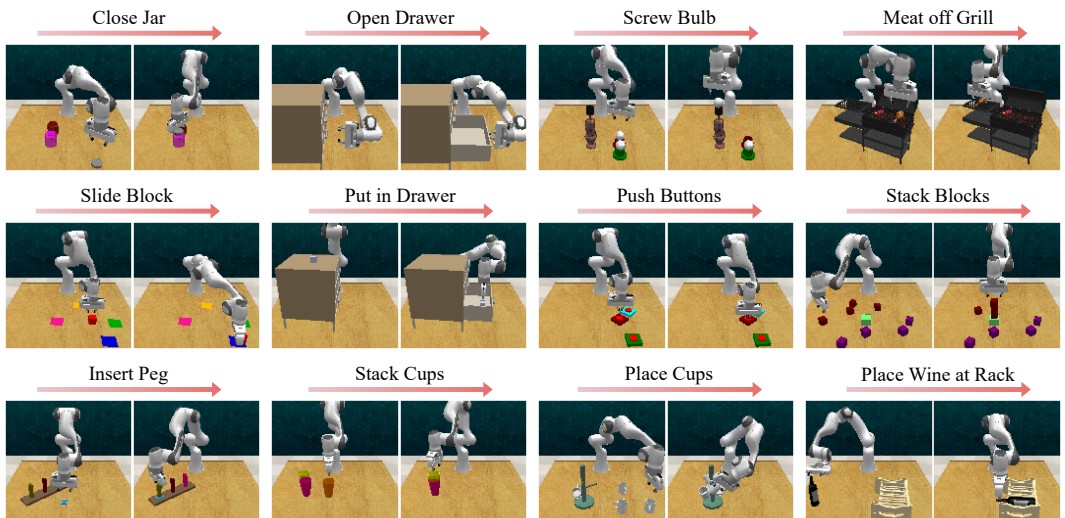

Figure 8: Visualization of 12 base tasks.

Table 4: The 12 base tasks selected from RLBench (James et al., 2020)

| Task | Variation Type | # of Variations | Avg. Keyposes | Language Template |
|------|---------------|-----------------|---------------|-------------------|
| close jar | color | 20 | 6.0 | "close the — jar" |
| open drawer | placement | 3 | 3.0 | "open the — drawer" |
| screw bulb | color | 20 | 7.0 | "screw in the — light bulb" |
| meat off grill | category | 2 | 5.0 | "take the — off the grill" |
| slide block | color | 4 | 4.7 | "slide the block to — target" |
| put in drawer | placement | 3 | 12.0 | "put the item in the — drawer" |
| push buttons | color | 50 | 3.8 | "push the — button, [then the — button]" |
| stack blocks | color, count | 60 | 14.6 | "stack — — blocks" |
| insert peg | color | 20 | 5.0 | "put the ring on the — spoke" |
| stack cups | color | 20 | 10.0 | "stack the other cups on top of the — cup" |
| place cups | count | 3 | 11.5 | "place — cups on the cup holder" |
| place wine | count | 3 | 5.0 | "stack the wine bottle to the — of the rack" |

### B.1.1 CLOSE JAR

**Task:** Close the jar by placing the lid on the jar.
**filename:** close_jar.py
**Modified:** The modified success condition registers a single DetectedCondition to check if the jar lid is correctly placed on the jar using a proximity sensor, discarding the previous condition of checking if nothing is grasped by the gripper.
**Success Metric:** The jar lid is successfully placed on the jar as detected by the proximity sensor.

### B.1.2 OPEN DRAWER

**Task:** Open the drawer by gripping the handle and pulling it open.
**filename:** open_drawer.py
**Modified:** The `cam_over_shoulder_left` camera's position and orientation were modified to better observe the drawer. The camera was repositioned to [0.2, 0.90, 1.10] and reoriented to [0.5*math.pi, 0, 0].
**Success Metric:** The drawer is successfully opened to the desired position as detected by the joint condition on the drawer's joint.

### B.1.3 SCREW BULB

**Task:** Screw in the light bulb by picking it up from the holder and placing it into the lamp.
**filename:** light_bulb_in.py

**Modified:** No.
**Success Metric:** The light bulb is successfully screwed into the lamp and detected by the proximity sensor.

### B.1.4   MEAT OFF GRILL

**Task:** Take the specified meat off the grill and place it next to the grill.
**filename:** meat_off_grill.py
**Modified:** The `cam_over_shoulder_right` camera's position and orientation were modified to better observe the drawer. The camera was repositioned to [0.20,-0.36,1.85] and reoriented to [-0.85*math.pi, 0, math.pi].
**Success Metric:** The specified meat is successfully removed from the grill and detected by the proximity sensor.

### B.1.5   SLIDE BLOCK

**Task:** Slide the block to the target of a specified color.
**filename:** slide_block_to_color_target.py
**Modified:** No.
**Success Metric:** The block is successfully detected on top of the target color as indicated by the proximity sensor.

### B.1.6   PUT IN DRAWER

**Task:** Put the item in the specified drawer.
**filename:** put_item_in_drawer.py
**Modified:** The `cam_over_shoulder_left` camera's position and orientation were modified to better observe the drawer. The camera was repositioned to [0.2, 0.90, 1.15] and reoriented to [0.5*math.pi, 0, 0].
**Success Metric:** The item is successfully placed in the drawer as detected by the proximity sensor.

### B.1.7   PUSH BUTTONS

**Task:** Press the buttons of the specified color in order
**filename:** push_buttons.py
**Modified:** No.
**Success Metric:** The buttons are successfully pushed in order.

### B.1.8   STACK BLOCKS

**Task:** Stack a specified number of blocks of the same color in a vertical stack.
**filename:** stack_blocks.py
**Modified:** No.
**Success Metric:** The blocks are successfully stacked according to the specified color and number.

### B.1.9   INSERT PEG

**Task:** Insert a square ring onto the spoke with the specified color.
**filename:** insert_onto_square_peg.py
**Modified:** No.
**Success Metric:** The square ring is successfully placed onto the correctly colored spoke.

### B.1.10   STACK CUPS

**Task:** Stack two cups on top of the cup with the specified color.
**filename:** stack_cups.py
**Modified:** No.
**Success Metric:** The cups are successfully stacked with the correct cup as the base.

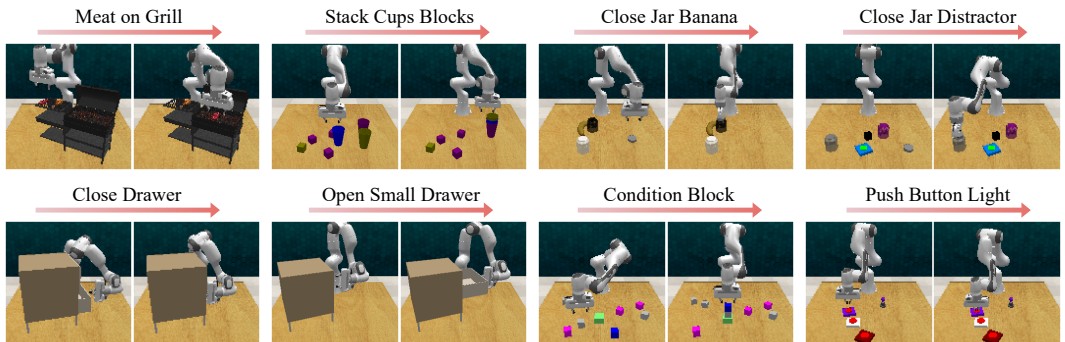

Figure 9: Visualization of 8 novel tasks.

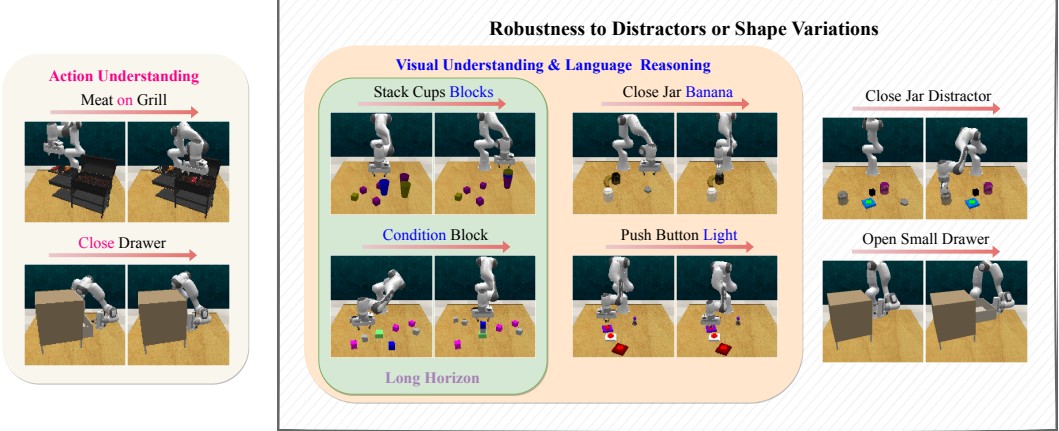

Figure 10: Three Novelty Categories for the Novel Tasks.

### B.1.11 PLACE CUPS

**Task:** Place a specified number of cups onto a cup holder.
**filename:** place_cups.py
**Modified:** No.
**Success Metric:** The cups are successfully placed onto the holder according to the task instructions.

### B.1.12 PLACE WINE AT RACK LOCATION

**Task:** Place the wine bottle onto the specified location on the wine rack.
**filename:** place_wine_at_rack_location.py
**Modified:** No.
**Success Metric:** The wine bottle is successfully placed at the correct rack location and released from the gripper.

### B.2 NOVEL TASK

As shown in Fig. 9, we create 8 novel tasks that differ from the original training tasks to test policy generalization. These tasks feature scenes and objects similar to those in the training tasks. We further define the novelty categories of the 8 novel tasks in our experiments to better explain the generalization improvements brought by GravMAD. As shown in Fig. 10, the designed novel tasks introduce three types of challenges to the model: **Action Understanding** (*meat on grill, close drawer*), **Visual Understanding & Language Reasoning** (*stack cups blocks, push buttons light, close jar banana, condition block*)—including two long-horizon tasks (*stack cups blocks* and *condition block*), and **Robustness to Distractors or Shape Variations** (*stack cups blocks, push buttons light, close jar banana, close jar distractor, open small drawer, condition block*).

Table 5: The 8 novel tasks changed based on base tasks.

| Task | Variation Type | # of Variations | Avg. Keyposes | Language Template |
|------|----------------|-----------------|---------------|-------------------|
| close drawer | placement | 3 | 2.0 | "close the — drawer" |
| close jar banana | placement | 2 | 6.0 | "close the jar closer to the banana" |
| close jar distractors | color | 20 | 6.0 | "close the — jar" |
| condition block | color, count | 72 | 11.0 | "build a tall tower out of — — cubes, and add a black block if it exists" |
| meat on grill | category | 2 | 5.0 | "put the — on the grill" |
| open small drawer | placement | 3 | 3.0 | "open the — drawer" |
| stack cups blocks | color | 20 | 10.0 | "Identify the most common color in the block pile, and stack the other cups on the cup that matches that color" |
| push button light | color | 20 | 2.0 | "push the button with the same color as the light" |

Specifically, **Action Understanding** refers to tasks involving changes in interaction actions with objects; **Visual Understanding & Language Reasoning** involve introducing entirely new operational rules or conditions compared to known tasks; and **Robustness to Distractors or Shape Variations** includes tasks that require interaction based on fixed object attributes (such as color, size, distance, or distractors). A summary of the seven tasks is provided in Table 5. We provide a detailed description of each novel task below and explain the modifications from the base tasks.

### B.2.1 MEAT ON GRILL

**Task:** Place either a chicken or a steak on the grill depending on the variation.
**filename:** meat_on_grill.py
**Base task:** meat off grill.
**Modified:** The task requires placing meat onto the grill, whereas the base task involves removing it. The `cam_over_shoulder_right` camera's position and orientation were modified to better observe the drawer. The camera was repositioned to [0.20,-0.36,1.85] and reoriented to [-0.85*math.pi, 0, math.pi].
**Success Metric:** The selected meat (chicken or steak) is successfully placed on the grill and released from the gripper.

### B.2.2 STACK CUPS BLOCKS

**Task:** Identify the most common color in the block pile, and stack the other cups on the cup that matches that color.
**filename:** stack_cups_blocks.py
**Base task:** Stack cups.
**Modified:** The task involves identifying the cup that matches the most common color among the distractor blocks, then stacking the other two cups on top. The base task is simply stacking the cups without considering block colors.
**Success Metric:** Success is measured when the correct cup is stacked with the other cups based on the color identification and all cups are within the target area defined by the proximity sensor.

### B.2.3 CLOSE JAR BANANA

**Task:** Close the jar that is closer to the banana by screwing on its lid.
**filename:** close_jar_banana.py
**Base task:** close jar.
**Modified:** The task involves identifying the jar closer to the banana and screwing its lid on, while the base task only requires closing a jar without proximity consideration.
**Success Metric:** The lid is successfully placed on the jar closest to the banana, confirmed by the proximity sensor.

### B.2.4 CLOSE JAR DISTRACTOR

**Task:** Close the jar by screwing on the lid, while distractor objects are present.
**filename:** close_jar_distractor.py
**Base task:** close jar.
**Modified:** The task includes distractor objects, such as a button and block, which are colored and placed near the jars. These objects have been encountered during training, adding complexity compared to the base task.

**Success Metric:** The jar lid is successfully placed on the target jar, confirmed by the proximity sensor.

### B.2.5 CLOSE DRAWER

**Task:** Close one of the drawers (bottom, middle, or top) by sliding it shut.
**filename:** close_drawer.py
**Base task:** open drawer.
**Modified:** The task involves closing the drawer instead of opening it.
**Success Metric:** The selected drawer is closed successfully, confirmed by the joint position of the drawer.

### B.2.6 OPEN DRAWER SMALL

**Task:** Open one of the smaller drawers (bottom, middle, or top) by sliding it open.
**filename:** open_drawer_small.py
**Base task:** open drawer.
**Modified:** The task involves opening a smaller drawer compared to the base task, with adjusted camera settings for better visibility.
**Success Metric:** The selected drawer is opened successfully, verified by the joint position of the drawer.

### B.2.7 CONDITION BLOCK

**Task:** Stack a specified number of blocks and, if the black block is present, add it to the stack.
**filename:** condition_block.py
**Base task:** stack blocks.
**Modified:** The task involves stacking a specified number of blocks, with an additional requirement to include the black block if it is present.
**Success Metric:** The correct number of target blocks are stacked, and if the black block is present, it is also correctly added to the stack.

### B.2.8 PUSH BUTTON LIGHT

**Task:** Push the button that matches the color of a light bulb on the first attempt.
**filename:** push_buttons_light.py
**Base task:** push button.
**Modified:** The task involves pressing a single button that matches the color of a light bulb. The button must be pressed correctly on the first attempt; repeated attempts are not allowed.
**Success Metric:** The correct button matching the light bulb's color is pressed on the first attempt.

### B.3 FAILURE CASES OF GRAVMAD

In this section, we analyze why GravMAD underperforms compared to the baseline model 3D Diffuser Actor on certain base tasks, particularly in the *"Place Wine"* task and drawer-related tasks.

As discussed in the main paper, GravMaps represent spatial relationships in 3D space, but this introduces a challenge: areas close to the sub-goal often share the same cost value, as seen in the value map on the right side of Fig. 11 (a). This uniform cost value can mislead the robot into assuming it should complete the sub-goal within that area. For tasks requiring precise actions, such as the *"Open Drawer"* task, GravMaps' coarse guidance may lead to suboptimal performance compared to 3D Diffuser Actor. In the left schematic of Fig. 11(a), the robot must grasp the center of a small handle to achieve optimal performance in the *"Open Drawer"* task. This high precision demand on the end-effector results in a lower success rate for GravMAD. This limitation extends to the *"Put in Drawer"* task, which depends on the successful completion of *"Open Drawer"*. Similarly, in the *"Place Wine"* task, insufficient predictive accuracy causes the robot to misalign the bottle with the correct slot by one unit, leading to failure.

In the VLM setting, sub-goal accuracy often suffers, as shown in Fig. 11(b), further reducing model performance. These inaccuracies typically arise from two factors: (1) SAM may fail to accurately

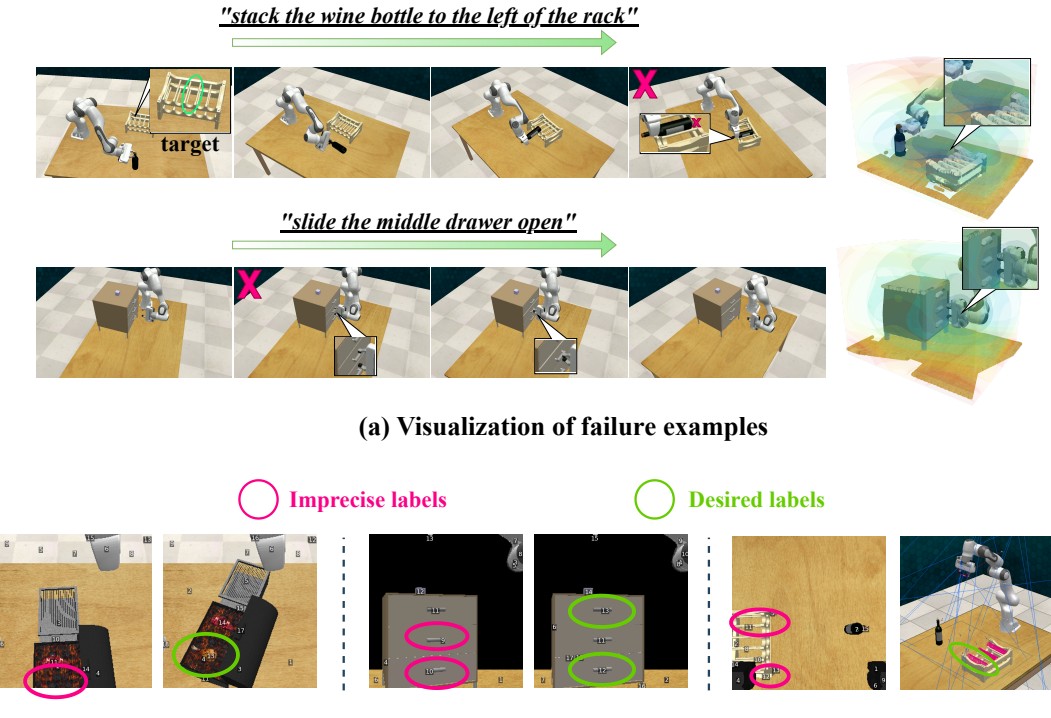

(a) Visualization of failure examples

(b) Comparison of imprecise labels and desired labels

Figure 11: **Failure cause analysis**, including (a) visualization of failure examples; (b) comparison of imprecise labels and expected labels.

identify ideal areas, leading to imprecise contextual information from the Detector module for tasks like *"Place Wine"*, *"Open Drawer"*, and *"Put in Drawer"*; (2) the camera's positioning may not capture the full scene, leaving some task-relevant objects out of view, as seen in tasks like *"Meat off Grill"*. To overcome these VLM limitations, potential solutions include: (1) integrating multi-view information into the Detector for a more comprehensive scene observation; and (2) using a more granular segmentation model to provide GPT-4 with a wider range of labels, improving the quality of the context generated by the Detector.

## C DISCUSSION

### C.1 THE RELATIONSHIP AND DIFFERENCES BETWEEN GRAVMAP AND VOXPOSER

The GravMap in GravMAD and the value maps in Voxposer (Huang et al., 2023) share the following connections and differences:

- **Number of value maps involved**: Voxposer utilizes multiple value maps, including the cost map, rotation map, gripper openness map, and velocity map. In our method, we only combine the cost map and gripper map, and their numerical values remain identical at this stage.

- **Structure and processing**: We further downsample the cost map and gripper openness map, transforming them into a point cloud structure containing position information and gripper states $(x, y, z, m_c, m_g)$, which we term GravMap. This sparse data structure not only efficiently represents sub-goals but also allows feature extraction using a point cloud encoder.

### C.2 The reason for not using the rotation map from Voxposer

GravMap does not currently use the rotation map from Voxposer because incorporating the rotation map could introduce significant distributional shifts between the guidance provided during the training and inference phases. During training, precise rotation guidance can be derived from expert trajectories. However, during inference, off-the-shelf foundation models often struggle to accurately interpret rotation information from visual and linguistic inputs, making it challenging to provide precise rotation guidance. To address this issue, future research will explore integrating rotation information from expert trajectories with object poses to generate few-shot prompts for off-the-shelf foundation models (Yin et al., 2024). This approach aims to enable LLMs to produce effective rotation guidance while reducing distributional shifts relative to the training data.

### C.3 Further Details on Sub-goal Keypose Discovery

#### C.3.1 Why sub-goals are extracted differently during training and inference

During the training phase of GravMAD, we use Sub-goal Keypose Discovery to extract sub-goals and generate GravMaps based on them. In contrast, during the inference phase, sub-goals are inferred by foundation models to generate GravMaps. The reasons for adopting different methods to generate GravMaps during the training and inference phases are as follows:

- **Efficiency and reliability during training:** Using Sub-goal Keypose Discovery to extract sub-goals during training is both simple and efficient. If foundation models were directly used to generate GravMaps as guidance during training, while they can indeed produce GravMaps, the results are generally coarser, less precise, and slower compared to expert trajectories. For example, due to limitations such as camera resolution or angles, foundation models may fail to fully observe the scene in some cases, leading to inaccurate sub-goal positions (failure cases are discussed in Appendix B.3). Under such circumstances, the quality of the training data cannot be guaranteed. Additionally, using foundation models to process large-scale data is practically infeasible due to their slow processing speed.

- **Simplifying the problem by avoiding semantic reasoning:** Extracting sub-goals from expert trajectories focuses solely on analyzing the robot's actions, thereby avoiding the complexity of semantic understanding and reasoning. Our key insight is that in task trajectories, certain actions in expert trajectories inherently carry semantic information (i.e., sub-goals, which may involve direct interactions with objects). These actions often exhibit distinctive features, such as the opening and closing of the gripper. The Keypose Discovery method (James & Davison, 2022) has already performed an initial filtering of these key actions, narrowing the scope for sub-goal selection. Based on this, we can quickly identify sub-goals through heuristic methods, which are also effective for long-horizon tasks.

It is worth noting that using different sub-goal generation methods during the training and inference phases may lead to a distributional shift. This occurs because the sub-goals generated by foundation models during inference are often less precise compared to those derived from expert trajectories, resulting in a discrepancy between the distributions of the training and inference phases. To address this issue, we apply data augmentation to the precise sub-goals generated from expert trajectories during the training phase. Specifically, as described in Line 279 of Algorithm 1, we introduce random offsets to the sub-goals generated during training (this processing is not applied to sub-goals generated during inference) and then generate GravMaps based on these perturbed sub-goals. This approach effectively reduces the risk of distributional shift to a certain extent.

#### C.3.2 Why use Sub-goal Keypose Discovery to filter keyposes

The Sub-goal Keypose Discovery method is essential for GravMAD because the original keyposes include both sub-goal keyposes and the intermediate steps required to achieve these sub-goals. These intermediate steps may involve precise alignment of the robotic arm with objects. However, foundation models often struggle to generate these intermediate steps, and even if they can, the results may exhibit significant distributional shifts compared to the guidance provided during the training phase. Additionally, generating only sub-goals reduces the complexity and difficulty of task reasoning for the foundation model while also simplifying the prompt engineering.

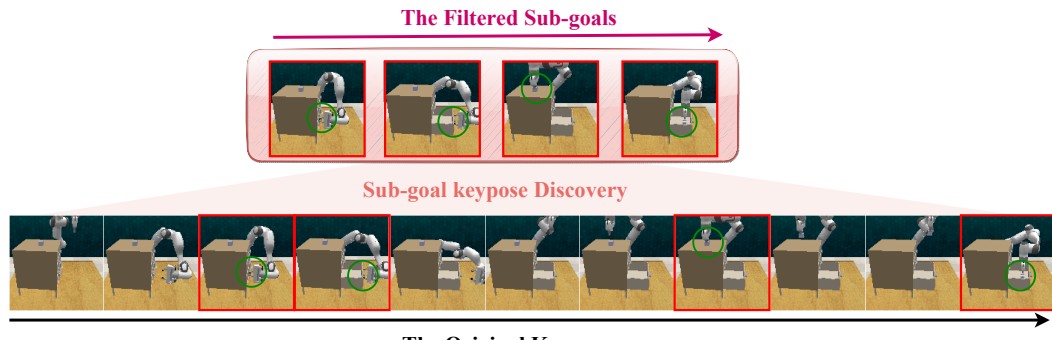

Figure 12: A comparison between the original keyposes and the filtered keyposes in the long-horizon task *put item in drawer*.

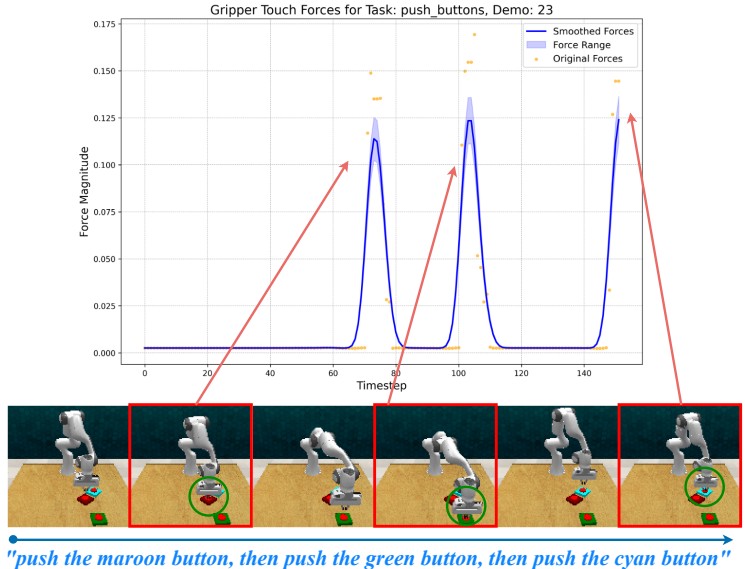

Figure 13: Visualization of sub-goal keypose discovery determining significant changes in **gripper_torch_force** during the *push button* task.

As shown in Fig. 12, for the long-horizon task *put item in drawer*, if only traditional keypose discovery methods are used, the extracted sub-goal stages would include **11 stages**. In contrast, when using our Sub-goal Keypose Discovery, the filtered sub-goals are reduced to just **4 stages**, perfectly aligning with the most critical phases of the task. This significantly reduces model inference time and improves task execution efficiency.

### C.3.3 CRITERIA FOR "SIGNIFICANT CHANGES" IN SUB-GOAL KEYPOSE DISCOVERY

To clearly explain the specific criteria for "significant changes" in our Sub-goal Keypose Discovery method, we visualized the changes in **gripper_touch_force** using the *push buttons* task as an example. As shown in Fig. 13, when the button is pressed, the gripper_touch_force value increases from nearly 0 to $0.1 \sim 0.15$. As the robotic arm lifts, the gripper_touch_force returns to 0. By analyzing these force changes, we can intuitively identify the sub-goal frames.

| | Voxposer | Act3D | 3D Diffuser Actor | **GravMAD (ours)** |
|---|---|---|---|---|
| Avg. Inference Time for Keypose Prediction (secs) | / | 0.04 | 1.78 | 1.81 |
| Avg. Task Completion Time (secs) | 448.01 | 14.08 | 49.45 | 97.04 |
| Avg. Inference Time per Sub-task Stage (secs) | 90.47 | / | / | 40.64 |

Table 6: Comparison of Inference Times.

| Models | Avg. Success ↑ | Avg. Rank ↓ | Close Jar | Open Drawer | Meat off Grill | Slide Block | Put in Drawer |
|---|---|---|---|---|---|---|---|
| Voxposer | 15.11 | 5.88 | 12.00 | 10.67 | 45.33 | 0.00 | 0.00 |
| Voxposer (Manual) | 22.06 | 4.92 | 13.33 | 18.67 | 69.33 | 0.00 | 0.00 |
| ChainedDiffuser (Oracle) | 29.72 | 4.42 | 82.67 | 0.00 | 52.00 | 2.67 | 0.00 |
| Act3D | 34.11 | 5.38 | 61.33 | 41.33 | 60.00 | 78.67 | 49.33 |
| 3D Diffuser Actor | 55.81 | 3.00 | 66.67 | 88.00 | 88.00 | 84.00 | 94.67 |
| GravMAD (Manual) | **69.17** | 1.63 | **100.00** | 76.67 | **89.33** | 93.33 | 78.67 |
| GravMAD (VLM) | 56.72 | 2.79 | **100.00** | 58.67 | 70.67 | 80.00 | 61.33 |
| Models | Push Buttons | Stack Blocks | Place Cups | Place Wine | Screw Bulb | Insert Peg | Stack Cups |
| Voxposer | 80.00 | 16.00 | 6.67 | 5.33 | 4.00 | 0.00 | 1.33 |
| Voxposer (Manual) | 86.67 | 36.67 | 13.33 | 10.67 | 6.67 | 0.00 | 9.33 |
| ChainedDiffuser (Oracle) | 62.67 | 15.00 | 22.33 | 48.67 | 25.33 | 4.00 | 41.33 |
| Act3D | 66.67 | 0.00 | 0.00 | 45.33 | 6.67 | 0.00 | 0.00 |
| 3D Diffuser Actor | 94.67 | 13.67 | 5.33 | 82.67 | 29.33 | 2.67 | 20.00 |
| GravMAD (Manual) | **98.67** | **56.67** | 5.33 | 77.33 | **66.67** | **32.00** | **57.33** |
| GravMAD (VLM) | 97.33 | 51.33 | 5.33 | 33.33 | 54.67 | 18.67 | 49.33 |

Table 7: Additional Multi-task test results on 12 base tasks.

| Models | Avg. Success ↑ | Avg. Rank ↓ | Close Drawer | Close Jar Banana | Close Jar Distractor | Condition Block | Meat On Grill | Open Drawer Small | Stack cups blocks | Push Buttons Light |
|---|---|---|---|---|---|---|---|---|---|---|
| Voxposer (Huang et al., 2023) | 34.29 | 3.25 | 96.00 | 17.33 | 22.67 | 25.00 | 38.67 | 6.67 | 0.00 | 68.00 |
| ChainedDiffuser (Oracle) (Xian et al., 2023) | 43.22 | 2.75 | 84.33 | 82.67 | 85.00 | 48.00 | 29.00 | 0.00 | 41.33 | 30.00 |
| Act3D (Gervet et al., 2023) | 17.83 | 4.25 | 66.67 | 29.33 | 41.33 | 0.00 | 1.33 | 2.67 | 0.00 | 1.33 |
| 3D Diffuser Actor (Ke et al., 2024) | 29.38 | 3.375 | 81.33 | 48.00 | 42.67 | 27.00 | 0.00 | 2.67 | 2.67 | 30.67 |
| GravMAD (VLM) | **62.92** | 1.125 | **97.33** | **84.00** | **86.67** | **74.00** | **45.33** | **21.33** | 18.67 | **76.00** |

Table 8: Additional generalization results on 8 novel tasks.

# D    ADDITIONAL EXPERIMENTAL RESULTS

## D.1    INFERENCE TIME

We test the inference time of all models under the setting of 8 novel tasks using a single NVIDIA 4090 GPU. The results, shown in Table 6 (in seconds), indicate the following: models like Act3D and 3D Diffuser Actor, which do not rely on foundation model inference, have shorter inference times but lower success rates. In contrast, Voxposer spends a significant amount of time synthesizing trajectories. Our GravMAD requires more time than Act3D and 3D Diffuser Actor because it waits for the foundation model to process information and infer sub-goals for sub-tasks.

## D.2    ADDITIONAL BASELINE EXPERIMENTS

We introduce two additional baseline methods for performance comparison: Voxposer (Manual) and Chained Diffuser (Xian et al., 2023) (Oracle). Voxposer (Manual) means that we manually provide ground truth object pose information to Voxposer instead of relying on the inference results of the foundation model. In Chained Diffuser (Oracle), we provide the ideal position for each keypose, with the connections between keyposes generated using the local trajectory diffuser module from Chained Diffuser. The performance comparisons of these two baseline methods on 12 base tasks and 8 novel tasks are shown in Table 7 and Table 8, respectively.

From the experimental results, we observe the following:

- In the base task setting, Voxposer (Manual) shows a slight performance improvement when provided with ground truth object information but still falls short compared to our GravMAD (Manual).

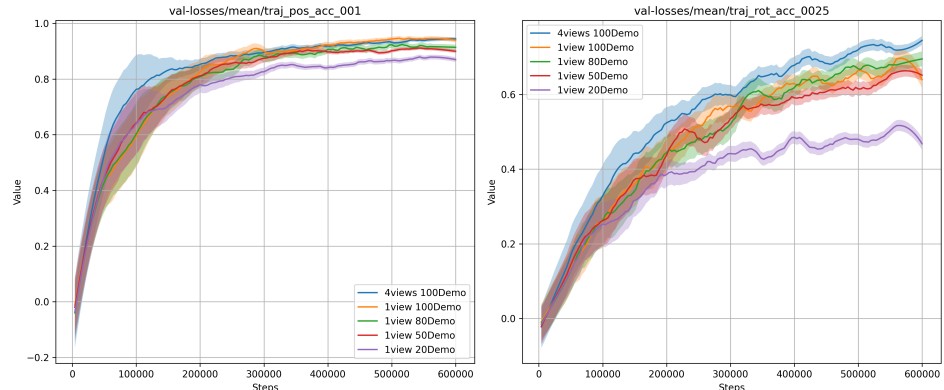

Figure 14: Comparison of validation curves under varying viewpoints and data sizes.

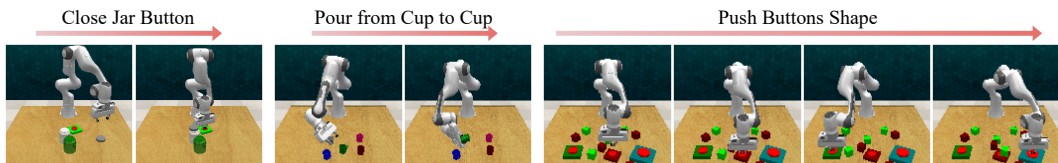

Figure 15: Visualization of additional novel tasks.

- For Chained Diffuser (Oracle), the keyposes come from ideal waypoints predefined in simulation, and the model effectively connects these keyposes, achieving a high success rate. However, in real-world scenarios, manually providing each keypose is impractical. Even with precise keyposes, Chained Diffuser (Oracle) still performs worse than our GravMAD (VLM).

### D.3 SCALABILITY OF GRAVMAD

To evaluate the scalability of our proposed method with respect to data volume, we conduct training comparisons using five different demonstration dataset sizes and visualize the corresponding validation curves. The experimental results are presented in Fig. 14, with the validation curves reflecting two key metrics:

1) The proportion of predicted positions in the validation set with an error less than 0.01 (left subplot in Fig. 14).

2) The proportion of predicted rotations in the validation set with an error less than 0.025 (right subplot in Fig. 14).

The results in Fig. 14 clearly demonstrate that the model's performance improves as the number of expert demonstrations and the number of viewpoints increase. The key observations are as follows:

- With only 20 expert demonstrations, the model exhibits low overall performance, particularly in predicting rotation angles.

- Models trained with four viewpoints achieve significantly better performance, but this improvement comes at the cost of increased training time.

- As the number of expert demonstrations grows, the marginal improvement in model performance diminishes. This could be attributed to the model's parameter size not scaling proportionally with the increase in data volume.

These results highlight the benefits of larger datasets for enhancing model performance. However, they also underscore the need for further optimization in model architecture and resource allocation

| Task | Variation Type | # of Variations | Avg. Keyposes | Language Template |
|---|---|---|---|---|
| push button shape | color | 20 | 2.0 | "Press the buttons in order of their size, from smallest to largest" |
| button close jar | color | 20 | 8.0 | "after close the — jar, push the button" |
| pour from cup to cup | color | 20 | 6.0 | "pour liquid from the — cup to the — cup" |

Table 9: Description of Additional Novel Tasks.

| Additional Novel Task | Voxposer | Act3D | 3D Diffuser Actor | GravMAD (Ours) |
|---|---|---|---|---|
| Push Buttons Shape (*Difficult Task*) | 0 | 0 | 0 | **62.66** |
| Button Close Jar (*Combination of Skills*) | 0 | 0 | 0 | 0 |
| Pour From Cup to Cup (*Completely New*) | 0 | 0 | 0 | 0 |

Table 10: Generalization Performance Comparison on Additional Novel Tasks.

to effectively harness the potential of large-scale data. Without such improvements, the diminishing returns observed with increasing data may limit scalability in practical applications.

## D.4 ADDITIONAL NOVEL TASKS

We evaluate the performance of baseline methods and GravMAD on three additional novel tasks, with detailed descriptions provided in Table 9 and Fig. 15. These tasks include a highly challenging one (Push Buttons Shape), a task that requires integrating skills learned during training (Button Close Jar), and a task involving entirely new objects compared to the training set (Pour From Cup to Cup).

The results are presented in Table 10 . The "Push Buttons Shape" task evaluates the model's ability to handle long-horizon planning, language reasoning, and robustness to visual perturbations. Under these conditions, all baseline methods fail to complete the task, whereas GravMAD performs well, showcasing its potential for generalization. For the "Button Close Jar" task, the results indicate that GravMAD still struggles with long-horizon tasks requiring the integration of multiple skills. In the entirely new task "Pour From Cup to Cup", GravMAD successfully identifies task-relevant objects but fails to complete the task due to incorrect actions. This failure is likely caused by a significant mismatch between the training data and the test environment.

## D.5 ADDITIONAL ABLATION STUDY

To investigate the impact of the cost map on model performance, we perform more detailed experiments on the "w/o Cost map" ablation setting. In this ablation study, due to the inherent limitations of the encoder, the GravMap containing only the gripper map cannot be effectively processed. For instance, when the sub-goal requires the robotic arm to perform a "close everywhere" operation, $m_g$ becomes a zero structure. Such an $m_g$ cannot be properly parsed by the DP3 Encoder, resulting in gradient vanishing during the training process.

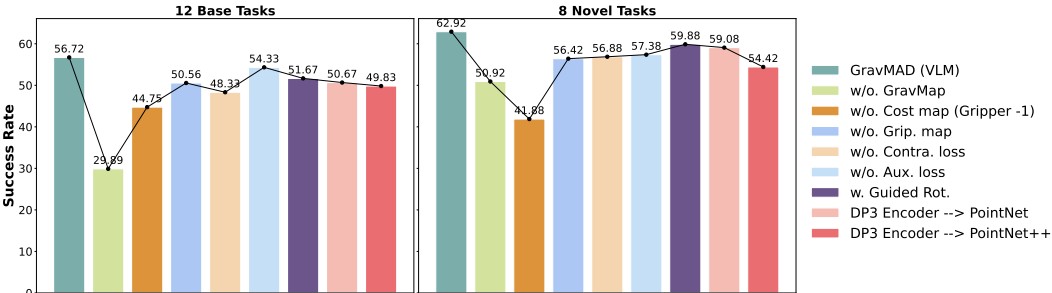

Figure 16: **Additional Ablation Studies**. We represent the gripper closure in the gripper map under "w/o. Cost map" as -1 instead of 0, enabling the encoder to correctly process this data structure.

To address this issue, we modify the gripper map in the "w/o Cost map" setting by changing the closed state representation from 0 to -1, enabling the encoder to correctly process this data structure. The experimental results are shown in Fig. 16. The results show that removing the cost map causes a significant performance drop compared to the original model: a decrease of 11.97% on 12 base tasks and 21.04% on 8 novel tasks. These findings clearly highlight the critical role of the cost map in ensuring the performance of the GravMAD model.

## D.6 REAL WORLD EVALUATION

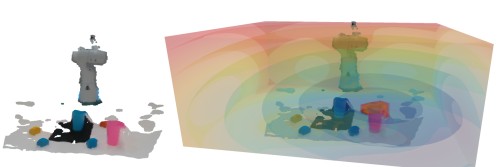 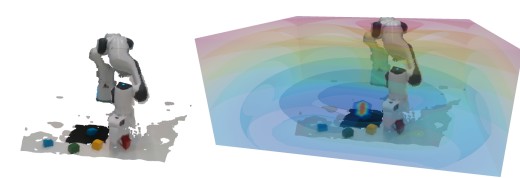

| Real-world Task | Open Drawer | Toy in Drawer | Mouse on Pad | Stack Cup | Stack Block Same |
|---|---|---|---|---|---|
| GravMAD (%) | 80 | 90 | 100 | 60 | 50 |
| Real-world Task | Place Cup | Stack Block | Stack Cup Blocks | Wired Mouse on Pad | Colored Toy in Drawer |
| GravMAD (%) | 10 | 40 | 40 | 100 | 70 |

Table 11: **Real-robot Results.** Success rates of GravMAD on 10 real-world tasks. These tasks include both manipulation and placement challenges. Above the table are the point clouds and GravMaps for Stack Cup Blocks and Stack Block, respectively.

We use a Franka Emika robot to validate Grav-MAD's multi-task generalization ability across 10 real-world tasks. Each task involves variations in placement, and some tasks include color variations. Compared to the base tasks, the novel tasks introduce new objects and new instructions. The base tasks include:

- **Open Drawer** (task description: open top drawer)
- **Place Cup** (task description: put the yellow toy in the top drawer)
- **Mouse on Pad** (task description: put the wireless mouse on pad)
- **Stack Cup** (task description: stack color1 cup on top of color2 cup)
- **Stack Block Same** (task description: stack blocks with the same color)
- **Place Cup** (task description: place one cup on the cup holder)

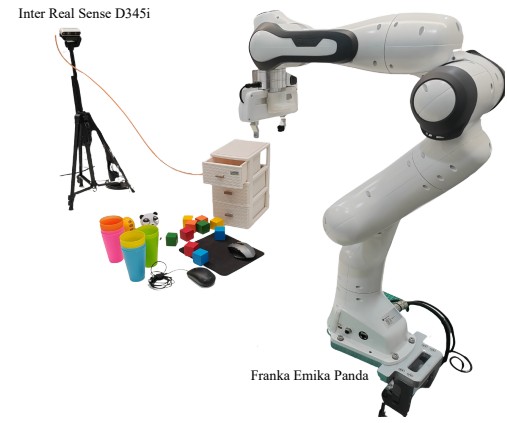

Figure 17: Real-Robot Setup with RealSense D435i and Franka Panda.

The novel tasks involve:

- **Stack Block** (task description: stack color1 block on top of color2 block)
- **Stack Cup Blocks** (task description: identify the most common color in the block pile, and stack the other cups on the cup that matches that color)
- **Wired Mouse on Pad** (task description: put the wired mouse on pad)
- **Colored Toy in Drawer** (task description: put the Black and white toy in the top drawer)

We position a RealSense D435i camera in front of the robot to capture images, which are downsampled from the original resolution of 1280×720 to 256×256, as shown in fig. 17. During training, we collect

20 demonstrations for each base task to train the model. During inference, similar to the simulation setup, GravMAD predicts the next keypose, and we use the BiRRT planner provided by MoveIt! ROS to guide the robot to reach the predicted keypose. For evaluation, we run 10 episodes for each task and report the success rate.

The inference performance of GravMAD on 6 base tasks and 4 novel tasks is shown in Table 11. These results demonstrate that GravMAD can effectively reason about 3D manipulation tasks in real-world robotic scenarios, leveraging associated visual information and generalizing to novel tasks. The video demonstrations are available at: `https://gravmad.github.io`

## E LIMITATIONS AND POTENTIAL SOLUTIONS

Despite GravMAD demonstrating strong generalization capabilities across the 3 categories and 8 novel tasks showcased, it still has certain limitations. The following section discusses some of the limitations not covered in the main text and their potential solutions:

- **Limitations of heuristic Sub-goal Keypose Discovery:** The current method relies on predefined heuristic rules, which may struggle to adapt to tasks with more complex or ambiguous sub-goal structures. Future research could explore more adaptive or learning-based strategies, such as incorporating diffusion models (Black et al., 2024) or generative models (Shridhar et al., 2024) to generate sub-goals, to further enhance the robustness and flexibility of the method.

- **Dependence on Detector accuracy and inference time:** The Detector's accuracy during the inference phase has a significant impact on the results, and its relatively long inference time remains a bottleneck. Future work could integrate observations from multiple viewpoints to provide a more comprehensive scene understanding and improve detection accuracy. Alternatively, more granular segmentation models could be leveraged to provide richer labels for foundation models, thereby improving the quality of the context generated by the Detector.

- **Limited guidance for end-effector orientation:** The current GravMap framework does not effectively guide the robot's end-effector orientation, limiting its applicability to tasks requiring precise orientation control. A potential improvement involves combining rotation information from expert trajectories with object poses to generate few-shot prompts for off-the-shelf foundation models (Yin et al., 2024). By leveraging such few-shot prompts, foundation models could produce more precise and effective rotation guidance.

- **Challenges in generalization:** While GravMAD performs exceptionally well on tasks similar to those seen during training, its generalization ability is still limited for tasks with significant differences from the training set, such as entirely unseen tasks or challenging tasks requiring a combination of multiple learned skills. Expanding GravMAD's capability to flexibly integrate multiple learned skills will be a key direction for future research. One feasible direction is to combine exploration-based learning with reinforcement learning (Hao et al., 2024).

- **Dependence on GravMap for Sub-goal Representation:** The GravMap framework relies on point cloud structures for sub-goal representation, which, while effective, may add unnecessary complexity in scenarios where simpler representations, such as a single point or relative coordinates, could suffice. The competitive performance of the "w/o GravMap" variant on novel tasks suggests that alternative representations could simplify the model without compromising performance. Defining sub-goals as relative coordinates with respect to the gripper's current position, leveraging proprioceptive information, is a promising direction. This approach could possibly introduce more data variation, enhance adaptability to spatial changes, handle imprecise sub-goals, and naturally encode directional information. Future research could explore this direction further to achieve a balance between simplicity and performance, potentially enhancing the generalization capability of the model while reducing reliance on GravMap.

By addressing these limitations, we anticipate that GravMAD will demonstrate stronger adaptability and practical value in more diverse tasks.

