# OpenReview forum: "GravMAD: Grounded Spatial Value Maps Guided Action Diffusion for Generalized 3D Manipulation"
_ICLR.cc/2025/Conference — ICLR 2025 Poster_

### Official Review · Reviewer_vbZp · 2024-10-21

**Soundness:** 3
**Presentation:** 2
**Contribution:** 2
**Rating:** 6
**Confidence:** 4

**Summary:**

This paper proposes a novel method for learning from demonstrations while incorporating knowledge from foundation models to tackle robotic manipulation. It learns to predict robot end-effector actions using a diffusion process and enhances the representation using 3D cost maps introduced in previous work. These cost maps are computed using heuristically extracted sub-goals during training, while foundation models are used instead at inference. In this way, the proposed method is able to combine strong generalisation capabilities from foundation models and learn precise robot actions from available demonstrations. The experiments show strong performance when compared to the considered baselines, while ablation studies provide valuable insights into different components of the method.

**Strengths:**

- The motivation of the paper is strong and well-explained.
- Figures (both in the main paper and the appendix) are clear and very helpful in understanding the proposed method.
- The technical challenge of integrating many components together in a unified framework is well-executed and appreciated.
- Ablation studies are well-executed and provide valuable insights into the method.
- Overall, utilising foundation models together with some demonstrations to achieve both, task generalisation and high precision seems like a great direction.

**Weaknesses:**

- Novelty and contribution claims are not very clear:
    - 1) Discovering sub-goals and predicting them to guide robot actions has been done before [1,2,3,4]. Is the novelty here mainly the 3D part combined with foundation models? Maybe the difference could be emphasised more.
    - 2) From the paper, it seems that GravMaps are Cost and Gripper openness maps used in VoxPoser, just renamed. Without properly explaining the difference, it shouldn’t be called a novel contribution.
    - 4) Claiming to design two types of tasks (line 124) and then using base tasks from RLBench for one type can come off as being disingenuous about the contribution claims. Rephrasing the 4th contribution to focus more on the evaluation rather than task design would be beneficial.
- Some parts of the method and design choices need to be better explained:
    - How GravMap tokens are used in the network architecture is not described in the main text of the paper.
    - Explaining in more detail why sub-goals are extracted differently during training and inference would greatly benefit the clarity of the paper. It becomes obvious while reading the paper, but initially, it can cause confusion.
    - It is not clear why rotation maps from VoxPoser are not used. Justifying it would make the overall approach clearer.
    - GravMap Synthesis during Inference is not explained clearly. Lines 305-309 are very dense with information, but it’s very difficult to understand how the sub-goals are actually extracted. If it is done just using existing methods, then contribution claims should be adjusted accordingly.
- It is not clear if the proposed sub-goal keypose discovery is actually needed. There are no experiments using keyposes, without filtering them. If there are other reasons to filter keyposes, they should be explained.
- Overall, sub-goal keypose discovery seems very tailored towards specific types of tasks. Limitations of it (and how they could be lifted) are not discussed.
- What are 3 random seeds used for? Are the models re-trained using different seeds or just to change object poses and the stochastic diffusion process? If it’s the latter, then it is the same as having 3x episodes and the reported statistics don’t hold a lot of value.
- GravMAD (Manual) seems like an unfair comparison to the baselines. Maybe providing the same ground truth object pose information to other methods would be more valuable for such comparison.
- Given that the proposed method uses sub-goals to guide the diffusion process, selected baselines are not the most appropriate. Better baselines would be ChainedDiffuser [3] and/or Hierarchical Diffusion Policy [4].
- There are some contradictory and not complete claims in the paper:
    - Line 144 - Described methods already leverage demonstration data.
    - Line 205 - Quaternions, although they have the double cover property, are continuous unless additional restrictions are imposed (e.g. by requiring that the scalar part w≥0).
    - Line 206 stated that 6D rotation representation is used, but line 510 states that the model can predict non-unit quaternions. How is this possible?
    - What are “3D tasks”? Tasks that involve 3D observations or 6D manipulation? It is not a very clear term.
- Limitations of the proposed method are only superficially discussed.

[1] Black, Kevin, et al. "Zero-shot robotic manipulation with pretrained image-editing diffusion models." arXiv preprint arXiv:2310.10639 (2023).

[2] Kang, Xuhui, Wenqian Ye, and Yen-Ling Kuo. "Imagined subgoals for hierarchical goal-conditioned policies." CoRL 2023 Workshop on Learning Effective Abstractions for Planning (LEAP). 2023.

[3] Xian, Zhou, et al. "Chaineddiffuser: Unifying trajectory diffusion and keypose prediction for robotic manipulation." 7th Annual Conference on Robot Learning. 2023.

[4] Ma, Xiao, et al. "Hierarchical Diffusion Policy for Kinematics-Aware Multi-Task Robotic Manipulation." Proceedings of the IEEE/CVF Conference on Computer Vision and Pattern Recognition. 2024.

**Questions:**

- Some are raised in the Weaknesses section.
- What is the difference between GravMaps and the cost maps described in VoxPoser? Is it the downsampling part?
- Why are rotation maps from VoxPoser not used?
- Do you have any intuition on how well your proposed method would scale with more data? Would Imitation Learning catch up at some point (with a lot of training data)?
- Could you expand a bit more about the limitations of your proposed method?
- GravMaps are just an overparameterization of sub-goals at a high level. The fact that the ablation variant (w/o. GravMap) performs so poorly is not very intuitive. Could you provide more details about this ablation variant and maybe you have some intuition about its performance?

**Details Of Ethics Concerns:**

No concerns.

---

> ### Author Response · Authors · 2024-11-20
> **Response to Reviewer vbZp**
>
> **General Reply:** We sincerely thank Reviewer vbZp for recognizing the strengths of our work, including the strong motivation of the proposed method, the integration of multiple components into a unified framework, and the clarity of the figures and ablation studies that provide valuable insights. We address the concerns and questions in detail below.
>
> > Novelty and contribution claims are not very clear:
> >
> > - 1. Discovering sub-goals and predicting them to guide robot actions has been done before [1,2,3,4]. Is the novelty here mainly the 3D part  combined with foundation models? Maybe the difference could be  emphasised more.
> > - 2. From the paper, it seems that GravMaps are Cost and Gripper openness maps used in VoxPoser, just renamed. Without properly explaining the  difference, it shouldn’t be called a novel contribution.
> > - 4. Claiming to design two types of tasks (line 124) and then using base tasks from RLBench for one type can come off as being disingenuous  about the contribution claims. Rephrasing the 4th contribution to focus  more on the evaluation rather than task design would be beneficial.
>
> **Response to W1:** Thank you for your suggestion. We acknowledge that the concept of task sub-goals has been explored in prior works. The primary innovation of GravMAD, however, lies in utilizing sub-goals as a mechanism to integrate foundation model-based approaches with imitation learning-based methods, rather than in proposing the concept of sub-goal discovery itself. This integration enables more effective task execution by combining the strengths of both approaches.
>
> ***To address your concern, we have revised the potentially misleading statements in the Introduction section of the paper. We have also cited the four papers mentioned by the reviewer (among which ChainedDiffuser was already cited) and explicitly acknowledged their contributions to the development and application of sub-goals in this context.*** Once again, we sincerely thank Reviewer vbZp for bringing this to our attention and helping us improve the clarity and accuracy of our work.
>
> **Response to W2:**  Thank you for your feedback. The differences between our GravMaps and the value maps used in Voxposer are primarily as follows:
>
> 1. **Number of value maps**: Voxposer employs multiple value maps, including the cost map, rotation map, gripper map, and velocity map. In contrast, our method focuses on combining only the cost map and gripper map, with their numerical values remaining identical at this stage.
> 2. **Structure and processing**: While Voxposer uses dense value maps, we downsample the cost map and gripper map into a point cloud structure that includes position and gripper information $(x, y, z, m_c, m_g)$. This structure, which we term GravMap, provides a more efficient and sparse representation of sub-goals. Additionally, GravMaps are compatible with point cloud encoders, enabling the extraction of rich spatial features and enhancing their utility for downstream tasks.
>
> ***To clarify these distinctions further, we have included a detailed discussion on the differences between GravMaps and the value maps used in Voxposer in Appendix C.1 of the revised manuscript.*** Thank you again for your valuable comment, which has helped us improve the clarity of our paper.
>
>
>
> **Response to W3:** Thank you for pointing out this issue. The 12 base tasks were indeed directly selected from RLBench rather than "designed" by us. ***We have revised the corresponding statements in the paper.***

---

> ### Author Response · Authors · 2024-11-20
> **Response to Reviewer vbZp (continued)**
>
> > Some parts of the method and design choices need to be better explained:
> >
> > - How GravMap tokens are used in the network architecture is not described in the main text of the paper.
> > - Explaining in more detail why sub-goals are extracted differently  during training and inference would greatly benefit the clarity of the  paper. It becomes obvious while reading the paper, but initially, it can cause confusion.
> > - It is not clear why rotation maps from VoxPoser are not used. Justifying it would make the overall approach clearer.
> > - GravMap Synthesis during Inference is not explained clearly. Lines  305-309 are very dense with information, but it’s very difficult to  understand how the sub-goals are actually extracted. If it is done just  using existing methods, then contribution claims should be adjusted  accordingly.
>
> **Response to W4:** Thank you for your comment. The usage of GravMap tokens in the network is explained in detail in Appendix A.3 and illustrated in Figure 5 of the original paper. ***To better address the reviewer’s concerns, we have included additional descriptions in Section 3.3 of the revised manuscript (highlighted in blue)***. While these revisions provide a concise explanation of how GravMap tokens are utilized, the full details, including step-by-step descriptions and illustrations, are available in Appendix A.3 for a more comprehensive understanding. We recommend referring to this section for deeper insights due to space constraints in the main paper.
>
> **Response to W5:** Thank you for the question. The reasons for using different methods to generate GravMaps during the training and inference phases are as follows: 1. Efficiency and reliability during training; 2. Simplifying the problem by avoiding semantic reasoning. ***We have added a detailed explanation of this issue in Appendix C.3.1 of the revised paper. Thank you again for your valuable suggestion.***
>
> **Response to W6:** The primary reason we currently do not use the rotation map from Voxposer is the significant distributional shift that may arise between the guidance provided during training and inference. During training, precise rotation guidance can be extracted directly from expert trajectories, ensuring high-quality supervision. However, during inference, off-the-shelf foundation models often struggle to accurately interpret 3D spatial relationships based on visual and linguistic inputs, which limits their ability to provide precise rotation guidance. This discrepancy between the high-quality training guidance and the less reliable inference guidance can lead to substantial performance degradation.
>
> To address this issue, our future research plans include combining rotation information from expert trajectories with object pose data to create few-shot prompts for off-the-shelf foundation models [1]. By leveraging these few-shot prompts, large language models (LLMs) could generate more accurate and effective rotation guidance, reducing the distributional shift from the training data and improving overall inference performance. ***We have added a detailed explanation of this issue in Appendix C.2 of the revised manuscript. Thank you again for highlighting this point and helping us improve the clarity of our work.***
>
> **Response to W7:**  Thank you for your comment. The synthesis process of GravMap during inference proceeds as follows: ***1) the robot's observations and task language instructions are processed by the Detector to produce the Context. 2) Based on this Context, the Planner and Composer generate Sub-goals, which are subsequently used to create the GravMap***. This inference process is depicted in Figure 2 of the main paper.
>
> Our approach adapts this process from the existing Voxposer method, with significant differences in the implementation. In Voxposer, tools like the open-vocab detector OWL-ViT, Segment Anything, and the video tracker XMEM are used to track object masks, which, combined with RGB-D observations, reconstruct the object/part point cloud as the Context. In contrast, our method relies solely on SoM-driven GPT-4 to interpret instructions and generate the Context. **This results in a more lightweight and efficient inference pipeline.**
>
> To provide further clarity, Appendix A.2.2 of the paper details the entire GravMap synthesis process during inference, including comprehensive descriptions of the Detector, Planner, and Composer. **Moreover, we have shared all inference prompts used with the foundation models on our anonymous project website for full reproducibility: https://gravmad.github.io.**
>
> We hope this explanation and the resources provided address your concerns. Thank you again for your valuable feedback.
>
> [1] Yin, Yida, et al. "In-Context Learning Enables Robot Action Prediction in LLMs." *arXiv preprint arXiv:2410.12782* (2024).

---

> > ### Author Response · Authors · 2024-11-20
> > **Response to Reviewer vbZp (continued)**
> >
> > > 1. It is not clear if the proposed sub-goal keypose discovery is  actually needed. There are no experiments using keyposes, without  filtering them. If there are other reasons to filter keyposes, they  should be explained.
> > >
> > > 2. Overall, sub-goal keypose discovery seems very tailored towards  specific types of tasks. Limitations of it (and how they could be  lifted) are not discussed.
> >
> > **Response to W8:**  Thank you for your comments. **The Sub-goal Keypose Discovery method is critical** because the original keyposes include both sub-goal keyposes and intermediate steps required to achieve these sub-goals. These intermediate steps, such as precise alignment of the robotic arm with objects, are often challenging for foundation models to generate accurately. Even when these models can produce intermediate steps, they frequently exhibit significant distributional shifts compared to the guidance provided during training, which can negatively impact performance.
> >
> > Moreover, focusing solely on sub-goals simplifies task reasoning for the language model, reducing complexity and improving overall efficiency. It also streamlines prompt design (prompt engineering), making the process more practical and less error-prone. ***To further address your comments, we have expanded our explanation in Appendix C.4 and included Figure 12. This figure uses the "put item in drawer" task to visually illustrate why Sub-goal Keypose Discovery is essential for filtering keyposes and generating accurate sub-goals.***
> >
> >
> >
> > **Response to W9:**  Thank you for your feedback. You are correct that our current Sub-goal Keypose Discovery method is specifically designed based on the characteristics of two types of manipulation tasks. This design rationale is detailed in our **Response to W1 for Reviewer SKBE**.
> >
> > ***To better address your comments, we have added Appendix E in the revised manuscript, which provides a detailed discussion of the current limitations of the Sub-goal Keypose Discovery method and outlines potential directions for future improvement.*** We believe this addition clarifies the scope of our method and acknowledges areas where further development is needed.
> >
> >
> > > 1. What are 3 random seeds used for? Are the models re-trained using  different seeds or just to change object poses and the stochastic  diffusion process? If it’s the latter, then it is the same as having 3x  episodes and the reported statistics don’t hold a lot of value.
> > >
> > > 2. GravMAD (Manual) seems like an unfair comparison to the baselines.  Maybe providing the same ground truth object pose information to other  methods would be more valuable for such comparison.
> > >
> > > 3. Given that the proposed method uses sub-goals to guide the diffusion process, selected baselines are not the most appropriate. Better  baselines would be ChainedDiffuser [3] and/or Hierarchical Diffusion  Policy [4].
> >
> > **Response to W10:** Thank you for your question. The three random seeds are primarily used to introduce variability in object poses and the stochastic diffusion process during evaluation. The models are not re-trained with different seeds but are evaluated using the same pre-trained model across all seeds. This setup is designed to test the model's robustness to different initial conditions and stochastic processes.
> >
> > We understand your concern that this setup might effectively increase the sample size. However, we use a standardized method to compute evaluation metrics by calculating the mean and standard deviation across random seeds, ensuring that the results reflect the model's performance consistency rather than simply benefiting from additional samples. This approach validates the model's generalization ability while maintaining statistical rigor. Moreover, this evaluation method has been widely adopted in previous works[2] [3], and we follow the same computation criteria.
> >
> > [2] 3d  diffuser actor: Policy diffusion with 3d scene representations.
> >
> > [3] Act3d: 3d feature field transformers for multi-task robotic manipulation.
> >
> > **Response to W11 & W12:** Thank you for your question. We added two baseline methods for additional performance comparison: **Voxposer (Manual)** and **Chained Diffuser (Oracle)**. Voxposer (Manual) involves manually providing ground truth object pose information to Voxposer instead of relying on the inference results from the foundation model. For Chained Diffuser (Oracle), we provide the ideal positions for each keypose, while the connections between keyposes are generated using the local trajectory diffuser module of Chained Diffuser. ***The performance comparisons of these two baseline methods on 12 base tasks and 8 novel tasks have been provided in Table 7 and Table 8 of Appendix D.2, respectively.***
> > It is important to note that we are unable to provide accurate object poses and task sub-goals for act3d and 3dda, as their modeling frameworks do not allow such operations.

---

> > > ### Author Response · Authors · 2024-11-20
> > > **Response to Reviewer vbZp (continued)**
> > >
> > > > There are some contradictory and not complete claims in the paper:
> > > >
> > > > - Line 144 - Described methods already leverage demonstration data.
> > > > - Line 205 - Quaternions, although they have the double cover  property, are continuous unless additional restrictions are imposed  (e.g. by requiring that the scalar part w≥0).
> > > > - Line 206 stated that 6D rotation representation is used, but line  510 states that the model can predict non-unit quaternions. How is this  possible?
> > > > - What are “3D tasks”? Tasks that involve 3D observations or 6D manipulation? It is not a very clear term.
> > >
> > > **Response to W13:** Thank you for pointing this out. Indeed, existing methods also leverage demonstration data to a certain extent. However, our method enhances the utilization of demonstration data by building upon 3D diffuser actor and integrating foundation models. This allows our approach to not only imitate demonstrations more effectively but also achieve better generalization on unseen tasks. ***In response to your comment, we have revised the manuscript to clarify this point and better highlight our contribution, as indicated in the blue-highlighted text in Section 2***. Specifically, we emphasize how our method improves upon existing approaches by combining the strengths of diffusion-based architectures and foundation models. Thank you for helping us refine this aspect!
> > >
> > > **Response to W14:** Thank you for raising this issue. You are correct that quaternions are inherently continuous unless additional constraints, such as requiring $w \geq 0$, are imposed. In our work, the primary motivation for using the 6D rotation representation is to avoid potential discontinuities arising from such constraints or from conversions between quaternions and other representations during training or inference. The 6D representation provides a smooth and continuous parameterization of rotation without ambiguity (as it directly maps to a subset of the rotation matrix). This eliminates potential issues related to quaternion normalization or double-cover properties during optimization. ***We have clarified this motivation further in the revised manuscript, as indicated in the blue-highlighted text in Section 3.1.*** Thank you for your valuable suggestion!
> > >
> > > **Response to W15:**  Thank you for raising this issue. We have provided a detailed explanation in our third response to Reviewer fAE6. In the **w/o Cost Map** ablation experiment, we observed that during training, the GravMap tokens became entirely NaN, leading to gradient explosion. During inference, the predicted 6D rotations were completely disordered. Consequently, the converted quaternion $xyzw$ was a non-unit quaternion, meaning it **did not satisfy** $x^2 + y^2 + z^2 + w^2 = 1$. This caused the robotic arm to fail to execute valid rotational actions, resulting in a success rate of zero. We sincerely thank Reviewer vbZp and Reviewer fAE6 for pointing out this issue and apologize for the earlier unclear explanation. ***In the revised manuscript, we have provided further clarification on this phenomenon in lines 509–512 (highlighted in blue).***
> > >
> > > **Response to W16:**  Thank you for your question. The term "3D tasks" is indeed unclear. The correct term should be **"3D manipulation tasks"**, and we have revised it in the manuscript. 3D manipulation tasks refer to tasks where a robot interacts with objects in three-dimensional space, typically involving a combination of 3D observations and six-degree-of-freedom (6DoF) manipulation. ***We have updated this term in the revised manuscript (highlighted in blue).*** Thank you again for pointing this out!
> > >
> > >
> > >
> > > >Limitations of the proposed method are only superficially discussed.
> > >
> > > Thank you for your feedback. ***To better address your comments, we have expanded the discussion of the method's limitations in Appendix E of the revised manuscript.*** The updated discussion includes:
> > >
> > > 1. **Limitations of heuristic Sub-goal Keypose Discovery**: The current method may struggle with tasks involving complex or ambiguous sub-goal structures. Future work will explore more adaptive or learning-based approaches.
> > >
> > > 2. **Dependence on Detector accuracy and inference time**: The Detector’s accuracy significantly impacts results, and its relatively long inference time is a bottleneck. Improving its efficiency is a key direction for future research.
> > >
> > > 3. **Limited guidance for end-effector orientation**: GravMap currently lacks rotational guidance, which limits its applicability to tasks requiring precise orientation control.
> > >
> > > 4. **Generalization challenges**: The method performs well on tasks similar to those in training but struggles with tasks that differ significantly. Enhancing generalization is a priority for future research.
> > >
> > > **We have also summarized potential directions for improvement in Appendix E to address these challenges.** Thank you again for your valuable feedback.

---

> ### Author Response · Authors · 2024-11-20
> **Response to Reviewer vbZp (continued)**
>
> > - What is the difference between GravMaps and the cost maps described in VoxPoser? Is it the downsampling part?
> > - Why are rotation maps from VoxPoser not used?
> > - Do you have any intuition on how well your proposed method would  scale with more data? Would Imitation Learning catch up at some point  (with a lot of training data)?
> > - Could you expand a bit more about the limitations of your proposed method?
> > - GravMaps are just an overparameterization of sub-goals at a high  level. The fact that the ablation variant (w/o. GravMap) performs so  poorly is not very intuitive. Could you provide more details about this  ablation variant and maybe you have some intuition about its  performance?
>
> **Response to Q1:**  Thank you for your question. We have explained  this problem in the above **Response to W2**, and the detailed explanation has been reflected in Appendix C.1 of the revised manuscript.
>
> **Response to Q2:**  Thank you for your question. We have explained  this problem in the above **Response to W6**, and the detailed explanation  has been reflected in Appendix C.2 of the revised manuscript.
>
> **Response to Q3:**  Thank you for your question. To address your concern about how well our proposed method scales with more data, we conducted training comparisons using five different demonstration dataset sizes and visualized the validation curves. ***The results and detailed analysis have been added to Appendix D.3, with the validation curves presented in Figure 14***. From the results, we observe that the model’s performance improves as the number of expert demonstrations and viewpoints increases, but the improvement plateaus with very large datasets. This may be because the model’s parameter size does not scale proportionally with the data volume. We hypothesize that explicitly modeling shared structures, such as sub-goals and sub-stages guided by GravMap-like representations, could be an effective way to continue scaling performance with increasing data.
>
> Regarding imitation learning, we believe that with sufficient high-quality data, traditional imitation learning methods may reduce the gap, but their lack of generalization beyond the training set may still limit their scalability to novel tasks. In contrast, our method leverages foundation models, which offer robust generalization capabilities that complement the strengths of imitation learning.
>
> We hope this provides insight into the scalability of our approach, and we thank you for raising this important point.
>
>
>
> **Response to Q4:**  Thank you for your question. **A more detailed analysis of our proposed method's limitations and potential future improvements has been provided in Appendix E of the revised manuscript.**
>
> **Response to Q5:**  Thank you for your question. In the **w/o GravMap** ablation, we replaced the GravMap with a single point and its corresponding gripper openness $(x, y, z, \text{openness})$ as input, removing the step of processing sub-goals into GravMaps. The key difference is that GravMaps represent sub-goals as regions (with uniform values near the sub-goal and increasing costs farther away), while the ablation uses only a single coordinate and openness value. The performance drop occurs because relying solely on a point makes the model overly sensitive to slight deviations, focusing on the exact point rather than the broader sub-goal region. GravMaps provide a robust representation, allowing the model to handle variations better. Moreover, GravMaps, as point cloud structures, are encoded into dense features by the point cloud encoder, offering stronger 3D sub-goal representations and improving overall performance.
>
> ***To better address your concern, we have revised Section 4.4 of the manuscript to include an updated analysis of the w/o GravMap ablation experiment***. We hope this explanation clarifies the role and importance of GravMaps.
>
>
> We appreciate Reviewer vbZp' constructive feedback and will incorporate all suggestions to improve the clarity, completeness, and rigor of the manuscript. Thank you for helping us strengthen our work!

---

> > ### Comment · Reviewer_vbZp · 2024-11-22
> > **Response to the rebuttal**
> >
> > Thank you for your in-depth responses and additional experiments.
> >
> > Regarding the ablation variant "w/o GravMap", the results still seem very strange to me. When sub-goals are defined, they could be expressed as a relative position from the current position of the gripper (you have access to proprioception information) rather than absolute ones. This would introduce more variation in the data and would, and would increase generalization to spatial variations. Maybe this would completely solve this issue and would prove that GravMaps are not actually needed. I understand that now it might be outside the scope of this work to run such an experiment, but it is worth discussing it.
> >
> > Regarding the ablation variant "w/o Cost map". It is interesting to see that it can cause exploding gradients. Maybe I misunderstood, but doesn't removing cost maps result in just a 3D Diffusor Actor?

---

> > > ### Author Response · Authors · 2024-11-22
> > > **Response to Reviewer vbZp**
> > >
> > > Thank you for your positive feedback on our previous responses! Regarding your further questions, our replies are as follows:
> > >
> > > - **Regarding further discussion on the ablation variant "w/o GravMap”**
> > >
> > > Thank you for your questions and suggestions! Our current understanding is that your question pertains to "whether it is possible to replace the absolute position with a relative position in the "w/o GravMap” variant." Please let us know if we have misunderstood your question!
> > >
> > > Your idea has provided significant inspiration for our research! According to our experimental results, the "w/o GravMap” variant performs better than baseline methods on novel tasks, indicating that using a single point (instead of GravMap) as a sub-goal representation is feasible. Additionally, as you pointed out, defining sub-goals as relative coordinates with respect to the gripper’s current position can indeed introduce more data variation. Moreover, this representation seems to naturally encode directional information, offering an intriguing perspective for sub-goal representation.
> > >
> > > However, our experiments also demonstrate that point cloud-based sub-goal representations cover a broader range of target regions compared to a single point. This is particularly evident in the base tasks. When using only a single point as a sub-goal representation, the imprecision of the point and the model’s sensitivity to it may interfere with the final predictions, resulting in performance inferior to baseline methods. This further highlights the importance of GravMap in providing robust guidance and improving task success rates.
> > >
> > > We fully agree that this topic is highly meaningful for discussion. Furthermore, we believe that sub-goal representations based on relative coordinates present a promising direction for future exploration. ***To better address your suggestion, we have included an in-depth discussion of this issue in Appendix E of the revised manuscript.***
> > >
> > > That said, this research direction lies beyond the scope of our current work. The focus of this study is to validate the role of GravMap in achieving multi-task generalization. Representing sub-goals as relative coordinates would require significant adjustments to data representation and training setup. Due to time constraints, we are unable to conduct the suggested experiments during this submission cycle, but we plan to explore this direction further in future research.
> > >
> > > - **Regarding further discussion on the ablation variant "w/o Cost map"**
> > >
> > > Thank you for your question regarding the "w/o Cost map" ablation variant. The "w/o Cost map" variant means that GravMAD represents the sub-goal using only the Gripper map, and it is not merely a 3D Diffuser Actor. The Gripper map is a point cloud structure $(x, y, z, m_g)$, where $m_g$ encodes information about the gripper's open/close state. However, when the GravMap consists solely of the Gripper map, the DP3 encoder produces GravMap tokens with all values as `NaN`. These `NaN` values lead to exploding gradients, preventing the network from updating.
> > >
> > > To clarify this issue further, we printed the input and output values of the first linear layer in the DP3 encoder when processing the Gripper map:
> > >
> > > **Layer: Layer\_0, Input:**
> > >
> > > ```
> > > tensor([[[-0.1501, -0.5958,  0.6729,  0.0000],
> > >          [ 0.6546,  0.5122,  1.5165,  0.0000],
> > >          [ 0.1852,  0.1429,  0.8838,  0.0000],
> > >          ...,
> > >          [-0.1501,  0.0044,  0.8135,  0.0000],
> > >          [ 0.6546,  0.3737,  0.6729,  0.0000],
> > >          [ 0.3864,  0.1429,  0.7432,  0.0000]]], device='cuda:0')
> > > ```
> > >
> > > **Layer: Layer\_0, Output:**
> > >
> > > ```
> > > tensor([[[nan, nan, nan, ..., nan, nan, nan],
> > >          [nan, nan, nan, ..., nan, nan, nan],
> > >          [nan, nan, nan, ..., nan, nan, nan],
> > >          ...,
> > >          [nan, nan, nan, ..., nan, nan, nan],
> > >          [nan, nan, nan, ..., nan, nan, nan],
> > >          [nan, nan, nan, ..., nan, nan, nan]]], device='cuda:0')
> > > ```
> > >
> > > **Why do NaN values occur?**
> > >
> > > When the sub-goal specifies that the gripper should "close everywhere", $m_g$ becomes an all-zero structure. This oversimplifies the feature distribution, leading to the loss of effective gradients during activation. Consequently, the network's activations become `NaN`, causing exploding gradients and making it impossible to update the network.
> > >
> > > **We hope the above response addresses your concerns. Please feel free to reach out if you have any further questions!**

---

> > > > ### Comment · Reviewer_vbZp · 2024-11-22
> > > > **Response.**
> > > >
> > > > Thank you for your quick response and additional discussion.
> > > >
> > > > Regarding "w/o Cost map". If you are using an encoder that can not handle (by design) a specific input, the ablation becomes meaningless. You are trying to investigate the influence that the cost maps have on the performance, but you just show the inherent limitation of the encoder that you use.
> > > >
> > > > If I understand correctly, NaNs are the result of zero gradients. It is a common practice (which is actually quite beneficial) to normalise inputs between [-1, 1]. Thus if you represented the close gripper as -1 instead of 0, the gradients would still flow and there would be no NaN values in the network. Is that correct?
> > > >
> > > > Taking all of this into account, I would almost suggest removing this ablation, as again it does answer the question you are trying to answer. Unless you agree that representing gripper closing actions as -1 would solve this issue and you have the time to re-run your experiments (which is a big ask, I understand, and I don't want to imply that it is required).

---

> ### Author Response · Authors · 2024-11-22
> **Response to Reviewer vbZp**
>
> Thank you for your insightful suggestions and valuable feedback regarding the "w/o Cost map" ablation study.
>
> We have started re-training the "w/o Cost map" ablation model based on your suggestion to represent the gripper closing action as -1 instead of 0. Due to the lengthy training process, we estimate that the results will only be available toward the later stages of the rebuttal phase (approximately 30 hours are required to train on the 12 base tasks). **If this modification successfully resolves the NaN issue, we will include the updated experimental results in the revised manuscript. Conversely, if the issue persists, we will remove this ablation experiment from the paper, as per your suggestion, to ensure the clarity and scientific rigor of the manuscript.**
>
> We sincerely appreciate your constructive feedback, which has not only helped improve the quality of our work but also provided valuable insights for related research. If our response satisfactorily addresses your concerns, we earnestly and kindly request the reviewers to consider raising their scores and supporting the acceptance of this paper.

---

> > ### Comment · Reviewer_vbZp · 2024-11-24
> > **Response.**
> >
> > Thank you for your willingness to do additional experiments and improve the quality of the paper. I am happy to raise my score.
> >
> > Once the results of new experiments come in, please do share them and update the paper accordingly.

---

> ### Author Response · Authors · 2024-11-25
> **Response to Reviewer vbZp**
>
> Thank you for your valuable feedback and for raising your score. We greatly appreciate your recognition of our efforts to enhance the quality of the paper.
>
> **We have completed the new experiments and incorporated the results into Appendix D.5 of the revised manuscript. Additionally, we have updated Section 4.4 in the main paper to provide a clearer explanation of the performance of the "w/o Cost map" setting and included a reference to Appendix D.5.** The new experimental results, based on your suggestion, show that changing the gripper map's closed state representation from 0 to -1 in the "w/o Cost map" setting allows the encoder to correctly process this data structure. However, removing the cost map still results in a significant performance drop compared to the original model: a 11.97% decrease in testing performance on 12 base tasks and a 21.04% decrease on 8 novel tasks. These findings clearly demonstrate the critical role of the cost map in ensuring the performance of the GravMAD model.
>
> Thank you again for your constructive feedback and support!

---

### Official Review · Reviewer_SKBE · 2024-10-29

**Soundness:** 2
**Presentation:** 3
**Contribution:** 3
**Rating:** 6
**Confidence:** 3

**Summary:**

This manuscript investigates 3D manipulation tasks through robot learning. Traditional imitation learning excels with familiar tasks but struggles with new, unseen ones. Recent approaches use large foundation models to mitigate this but lack task-specific learning, often leading to failures in 3D environments. To overcome this, the author introduces a novel framework termed Grounded Space Value Map-guided Action Diffusion (GravMAD). Due to its promising results, I am tentatively inclined to accept the paper. However, I find certain aspects unclear, especially the evaluation of design choices and overall performance. I am open to revising my evaluation if the authors can address these concerns comprehensively and convincingly detail the contributions of their work.

**Strengths:**

1. GravMAD shows strong generalization capabilities, achieving at least 13.36% higher success rates than state-of-the-art baselines on 12 base tasks encountered during training and surpassing them by 28.63% on 8 novel tasks.
  2. The manuscript is well-written with a clear structure, logical flow, detailed technical descriptions, and intuitive chart presentations.

**Weaknesses:**

- Insufficient argumentation
  1. The manuscript mentions in section 3.2 that different sub-goal keypose discovery rules were designed for different types of manipulation tasks, which seems to lack a reasonable explanation and generalizability.
  2. The manuscript does not adequately justify the exclusion of GravMap’s guidance in rotation prediction, stating that “rotation actions are difficult to be guided explicitly through value maps.”
- Inadequate Evaluation:
  1.  Although it is claimed that this method is generalized, all experiments were conducted in the RLBench environment, and there is a lack of validation on other simulation environments or real robots.
  2. The main evaluation metric used is success rate, overlooking other important dimensions such as completion time, smoothness of action, etc.

**Questions:**

1. What are the specific criteria for “significant changes” in Sub-goal Keypose Discovery?
2. From Table 2, it is evident that the algorithm exhibits significant differences in performance on similar tasks, such as placing a cup and placing wine. Can you explain the reasons for this?

---

> ### Author Response · Authors · 2024-11-20
> **Response to Reviewer SKBE**
>
> **General Reply:** We sincerely thank Reviewer SKBE for recognizing the strengths of our work, including GravMAD's strong generalization capabilities, the clear structure and logical flow of the manuscript, and the intuitive presentation of technical details and charts. We address the concerns and questions in detail below.
>
> > Insufficient argumentation
> >
> > 1. The manuscript mentions in section 3.2 that different sub-goal  keypose discovery rules were designed for different types of  manipulation tasks, which seems to lack a reasonable explanation and  generalizability.
> > 2. The manuscript does not adequately justify the exclusion of  GravMap’s guidance in rotation prediction, stating that “rotation actions are difficult to be guided explicitly through value maps.”
>
> **Response to W1:** Thank you for your comment. Admittedly, our sub-goal keypose discovery rules are specifically designed based on the characteristics of two main types of manipulation tasks, as detailed in lines 262–266 of Section 3.2 of the main paper. These rules are tailored to grasping tasks and contact-based tasks, aligning with the unique properties of robotic manipulation. For grasping tasks, critical interaction moments are typically signaled by changes in the gripper’s contact force or open/close state, reflecting key stages of object manipulation. Conversely, in contact-based tasks (e.g., pushing buttons), where gripper open/close actions are not relevant, key interaction moments are instead indicated solely by changes in the gripper’s contact force.
>
> We acknowledge that this heuristic approach lacks generalizability and does not extend well to tasks involving tool use (**as noted in Appendix A.1**). However, it remains effective for the two task types discussed in the paper. Moving forward, we aim to address these limitations by incorporating learning-based approaches for sub-goal keypose discovery, which would enhance generalization and expand its applicability across a broader range of tasks.
>
> We sincerely thank you for your constructive feedback, which has guided us in identifying and prioritizing areas for improvement.
>
> **Response to W2:**  Thank you for your comment. The primary reason we currently do not use the rotation map from Voxposer is the significant distributional shift it may introduce between the guidance provided during the training and inference phases. During training, precise rotation guidance is directly derived from expert trajectories, ensuring accuracy and consistency. However, during inference, off-the-shelf foundation models often struggle to accurately interpret 3D spatial information based on visual and linguistic inputs, resulting in less reliable rotation predictions. This discrepancy can lead to performance degradation and inconsistent task execution.
>
> To address this limitation, our future research plans to explore combining rotation information from expert trajectories with object poses to generate few-shot prompts for off-the-shelf foundation models [1]. By leveraging these few-shot prompts, foundation models can produce more precise and effective rotation guidance, thereby reducing distributional shifts and improving inference performance.
>
> ***We have included a detailed explanation of this issue in Appendix C.2 of the revised manuscript. Thank you again for your valuable suggestion, which has helped us refine and clarify this aspect of our work.***
>
> [1] Yin, Yida, et al. "In-Context Learning Enables Robot Action Prediction in LLMs." *arXiv preprint arXiv:2410.12782* (2024).

---

> > ### Author Response · Authors · 2024-11-20
> > **Response to Reviewer SKBE (continued)**
> >
> > > Inadequate Evaluation:
> > >
> > > 1. Although it is claimed that this method is generalized, all  experiments were conducted in the RLBench environment, and there is a  lack of validation on other simulation environments or real robots.
> > > 2. The main evaluation metric used is success rate, overlooking other  important dimensions such as completion time, smoothness of action, etc.
> >
> > **Response to W3:**  Thank you for the suggestion regarding real-world experiments. ***In response, we have included the results of GravMAD on 10 real-world robotic manipulation tasks in Appendix D.5 of the revised manuscript. Additionally, videos showcasing these experiments have been made available on our project website: [https://gravmad.github.io](https://gravmad.github.io).***
> >
> > **Response to W4:** Thank you for your suggestion regarding additional evaluation metrics. To better address your concern, we have added an evaluation of the inference time for all models on the 8 novel tasks. ***The results are shown in the table below, and the related analysis has been included in Appendix D.1 of the revised manuscript.***
> >
> > |                                                   | Voxposer | Act3D | 3D Diffuser Actor | GravMAD(ours) |
> > | ------------------------------------------------- | -------- | ----- | ----------------- | ------------- |
> > | Avg. Inference Time for Keypose Prediction (secs) | /        | 0.04  | 1.78              | 1.81          |
> > | Avg.  Task Completion Time (secs)                 | 448.01   | 14.08 | 49.45             | 97.04         |
> > | Avg. Inference Time per Sub-task Stage (secs)     | 90.47    | /     | /                 | 40.64         |
> >
> > ---
> >
> > **Response to  Questions**
> >
> > > What are the specific criteria for “significant changes” in Sub-goal Keypose Discovery?
> >
> > **Response to Q1:** Thank you for your question. To address your concern and provide a clearer explanation of the specific criteria for "significant changes" in our Sub-goal Keypose Discovery method, we have visualized the variations in **gripper_touch_force** using the *push buttons* task as an example. ***These details have been included in Appendix C.3.3 and Figure 12 of the revised manuscript.***
> >
> > As illustrated in Figure 12 of Appendix C.3.3, when the button is pressed, the **gripper_touch_force** increases from nearly 0 to a range of 0.1–0.15. Once the robotic arm lifts, the **gripper_touch_force** returns to 0. By observing and analyzing these force changes, we can effectively and intuitively identify the key stages corresponding to sub-goal keyposes.
> >
> >
> >
> > > From Table 2, it is evident that the algorithm exhibits significant  differences in performance on similar tasks, such as placing a cup and  placing wine. Can you explain the reasons for this?
> >
> > **Response to Q2:** Thank you for your question. While the **place cups** and **place wine** tasks are both categorized as "place" tasks based on their names, they differ significantly in complexity and precision requirements. As shown in Figure 7 of Appendix B.1 in the paper, the **place cups** task involves picking up a cup and precisely aligning its handle with the rod of the cup rack to hang it securely. This process requires extremely high precision, as even a minor misalignment of the handle can lead to collisions and task failure.
> >
> > In contrast, the **place wine** task entails picking up a wine bottle and placing it on the wine rack. This task typically does not involve precise alignment or carry significant collision risks, making it comparatively easier to execute. These differences highlight the varying levels of difficulty between the two tasks, despite their similar categorization.
> >
> > We appreciate Reviewer SKBE' constructive feedback and will incorporate all suggestions to improve the clarity, completeness, and rigor of the manuscript. Thank you for helping us strengthen our work!

---

> > > ### Author Response · Authors · 2024-11-25
> > > **Response to Reviewer SKBE**
> > >
> > > Thank you once again for your time and effort in reviewing our paper and for providing such helpful comments! We have addressed the points you raised, with particular attention to clarifying our Sub-goal Keypose Discovery method (**Response to W1**), adding an explanation for not using the rotation map from Voxposer (**Response to W2**), presenting GravMAD's results on 10 real-world robotic manipulation tasks (**Response to W3**), evaluating the inference time for all models on the 8 novel tasks (**Response to W4**), providing a clearer explanation of the criteria for "significant changes" in our Sub-goal Keypose Discovery (**Response to Q1**), and further explaining the model's performance on the Place Cups and Place Wine tasks (**Response to Q2**). All corresponding clarifications have been incorporated into the revised version of the main paper and Appendix, with changes marked in blue.
> > >
> > > As the discussion stage between the authors and reviewers is approaching its conclusion, we kindly request the reviewers to review our revised paper and responses. **If our responses have adequately addressed your concerns, we would greatly appreciate it if you could reconsider your scores.**
> > >
> > > If the reviewers have any additional questions, we would be more than happy to provide further clarifications. We sincerely look forward to your feedback.

---

> > > > ### Author Response · Authors · 2024-11-27
> > > > **Response to Reviewer SKBE**
> > > >
> > > > We would like to express our sincere gratitude to you once again for your valuable feedback, which has played a crucial role in improving our manuscript and enhancing its competitiveness.
> > > >
> > > > **Although the discussion period has been extended, we would like to kindly remind you that the deadline for authors to upload the revised PDF is fast approaching.** If you have any remaining questions or concerns, we would greatly appreciate your prompt response. If you believe our responses have addressed your concerns, we would be grateful if you could consider revising our scores accordingly. We are happy to address any outstanding issues and are willing to make further revisions based on your feedback in the final stages!
> > > >
> > > > Thank you once again for your dedicated service to the academic community.

---

> ### Author Response · Authors · 2024-12-01
> **A Kind Reminder [Deadline Approaching]**
>
> Dear Reviewer SKBE,
>
> We would like to thank you once again for taking the time to review our work and provide feedback on our manuscript. As the extended discussion deadline (December 2) approaches, this is a gentle reminder to let us know if we have satisfactorily addressed the reviewer's concerns, focusing on: clarifying our Sub-goal Keypose Discovery method (Response to W1), adding an explanation for not using the rotation map from Voxposer (Response to W2), presenting GravMAD's results on 10 real-robot manipulation tasks (Response to W3), evaluating the inference time for all models on the 8 novel tasks (Response to W4), providing a clearer explanation of the criteria for "significant changes" in our Sub-goal Keypose Discovery (Response to Q1), and further explaining the model's performance on the Place Cups and Place Wine tasks (Response to Q2).  If you find it appropriate, we would be grateful if you could update your scores. We are happy to address any additional remaining concerns. We are grateful for your service to the community.
>
> Regards,
>
> The Authors of Submission 3039

---

> ### Comment · Reviewer_SKBE · 2024-12-03
> **Response to the rebuttal**
>
> Appreciations to the authors for their efforts. No further comments from my side.

---

> > ### Author Response · Authors · 2024-12-03
> >
> > Thank you for your thoughtful feedback and kind words. We sincerely appreciate the time and effort you have dedicated to reviewing our work. If you are satisfied with our responses and the revisions made, we kindly request you to consider updating the rate score or confidence score accordingly. Your support is greatly valued. Thank you!

---

### Official Review · Reviewer_fAE6 · 2024-10-29

**Soundness:** 2
**Presentation:** 3
**Contribution:** 2
**Rating:** 6
**Confidence:** 4

**Summary:**

This paper introduces GravMAD, a new approach that decomposes 3D manipulation tasks into key sub-goals, generates 3D spatial GravMaps to capture spatial relationships from these sub-goals, and employs an action diffusion policy to learn actions through imitation learning. This design aims to combine the precision of imitation learning with the generalization capabilities of foundation models. Extensive experiments conducted on RLBench show that GravMAD achieves improvements in both training tasks and novel tasks.

**Strengths:**

* The authors conduct extensive experiments, including comparisons with different types of baselines and a detailed ablation study.
* The authors provide diverse visualization in the paper and on the website, enhancing readers' understanding.
* The paper is well-organized and easy to comprehend.

**Weaknesses:**

* The novelty of the proposed method is limited, as it resembles a combination of previous works, Voxposer and 3D Diffuser Actor. Specifically, the authors use a similar approach to Voxposer by decomposing tasks and generating 3D spatial heatmaps for each sub-task, which are then integrated into the 3D Diffuser Actor architecture. The primary innovation appears limited to the proposed GravMAD framework.
* The comparison between GravMAD and 3D Diffuser Actor on training tasks indicates that, although the introduction of GravMap improves performance on most tasks, GravMAD's performance heavily relies on both the Detector and GravMap. If the Detector’s predictions lack precision, it can even degrade the method's performance, potentially compromising its robustness in scenarios where camera capture is unclear or ambiguous.
* In the ablation study w/o. Cost Map, it is surprising that the method achieves a zero score. Could the authors provide further explanation on why the predicted actions result in non-unit quaternions in this scenario? (This seems to conflict with the statement in Section 3.1, which mentions, "to avoid quaternion discontinuities, we use the 6D rotation representation.") Additionally, if the predicted quaternion is not-unit, we can normalize and convert it into a unit quternion.
* It is advisable to conduct real-world experiments, as many related works, such as Voxposer and 3D Diffuser Actor, also include real-world demonstrations.

**Questions:**

* The use of a foundation model, such as GPT-4o, during inference might be time-consuming and could potentially hinder real-time responsiveness in robotic manipulation. Could the authors provide the average inference time for GravMAD at each sub-task stage?
* During training, why not generate GravMap in the same way as during inference? (One possible reason may be that the foundation model’s inference time is too lengthy?) The GravMap generated from demonstration ground-truths may differ in distribution from the GravMap produced by the foundation model, which could impact inference performance.

---

> ### Author Response · Authors · 2024-11-20
> **Response to Reviewer fAE6**
>
> **General Reply:** We sincerely thank Reviewer fAE6 for recognizing the strengths of our work, including the extensive experiments with diverse baselines and a detailed ablation study, the visualizations that enhance understanding, and the paper's clarity and organization. We address the concerns and questions in detail below.
>
> > The novelty of the proposed method is limited, as it resembles a  combination of previous works, Voxposer and 3D Diffuser Actor.  Specifically, the authors use a similar approach to Voxposer by  decomposing tasks and generating 3D spatial heatmaps for each sub-task,  which are then integrated into the 3D Diffuser Actor architecture. The  primary innovation appears limited to the proposed GravMAD framework.
>
> **Response to W1:** Thank you for your feedback. While our method draws inspiration from Voxposer and 3D Diffuser Actor, GravMAD introduces three key innovations that go far beyond a simple combination of these approaches:
>
> 1. **GravMap as a Sub-goal Representation**: Unlike Voxposer, which uses voxel value maps for trajectory synthesis, GravMap acts as a crucial bridge between task decomposition and action planning, seamlessly integrating the understanding capabilities of the foundational model with the learning abilities from expert demonstrations into a unified framework.
> 2. **Sub-goal Keypose Discovery**: This novel mechanism efficiently identifies critical sub-goals in expert demonstrations, enabling the model to learn actions guided by sub-goals from expert demonstrations. This design allows the model to generalize to new tasks with high precision based on sub-goals during inference.
> 3. **Integrated Framework**: GravMAD integrates GravMap with the action diffusion process, creating a unified framework that balances the precision of imitation learning with the adaptability and generalization strengths of foundation models.
>
> ***Additionally, to address the reviewer’s concern, we have provided a detailed explanation of the relationship and differences between the GravMap used in GravMAD and the value maps in Voxposer in Appendix C.1 of the revised manuscript***. We hope this clarification resolves any uncertainties regarding this aspect. Once again, thank you for your valuable feedback, which has helped us improve our work.
>
>
>
> > The comparison between GravMAD and 3D Diffuser Actor on training tasks  indicates that, although the introduction of GravMap improves  performance on most tasks, GravMAD's performance heavily relies on both  the Detector and GravMap. If the Detector’s predictions lack precision,  it can even degrade the method's performance, potentially compromising  its robustness in scenarios where camera capture is unclear or  ambiguous.
>
> **Response to W2:**  Thank you for your comment. We recognize that the Detector and GravMap are key components of GravMAD, as illustrated in Fig. 2 of the main paper. The accuracy of the Detector is indeed crucial to our method's performance, as acknowledged in the analysis of GravMAD's failure cases in Appendix B.3. These cases demonstrate the limitations of the current Detector in certain scenarios.
>
> To address these challenges, we plan to:
>
> 1. **Integrate Multi-View Information**: Incorporating observations from multiple perspectives to provide a more comprehensive scene understanding and improve detection accuracy.
> 2. **Adopt a Granular Segmentation Model**: Leveraging more detailed segmentation to offer foundation models richer labels, improving the context quality generated by the Detector.
>
> ***To better reflect the reviewer's comments, we have incorporated the above-mentioned future improvement directions into Appendix E of the revised manuscript***. Thank you again for pointing out this important area for development！

---

> > ### Comment · Reviewer_fAE6 · 2024-11-25
> >
> > Thank you for your detailed response. After reviewing the revised paper, the other reviewers' comments, and the authors' responses, I find that most of my concerns have been addressed. Consequently, I have updated my score.

---

> > > ### Author Response · Authors · 2024-11-26
> > >
> > > We are delighted to hear that our response and the revisions to the manuscript have addressed your concerns, and we greatly appreciate your decision to raise your score. Thank you once again for your time and valuable feedback!

---

> ### Author Response · Authors · 2024-11-20
> **Response to Reviewer fAE6 (continued)**
>
> > In the ablation study w/o. Cost Map, it is surprising that the method  achieves a zero score. Could the authors provide further explanation on  why the predicted actions result in non-unit quaternions in this  scenario? (This seems to conflict with the statement in Section 3.1,  which mentions, "to avoid quaternion discontinuities, we use the 6D  rotation representation.") Additionally, if the predicted quaternion is  not-unit, we can normalize and convert it into a unit quternion.
>
> **Response to W3:**  Thank you for your feedback. We would like to clarify the relationship between quaternions and 6D rotation representations. These two concepts are not contradictory. As explained in Section 3.1 of the main paper, our model predicts a 6D rotation representation to avoid the discontinuities associated with quaternion prediction. During action execution, the predicted 6D rotation is converted into a quaternion $xyzw$, where $xyz$ represents the vector part defining the rotation axis as a unit vector, and $w$ is the scalar part corresponding to the cosine of the rotation angle. For valid rotations, the resulting quaternion is expected to be a unit quaternion, satisfying the constraint $x^2 + y^2 + z^2 + w^2 = 1$.
>
> In our **w/o Cost Map** ablation experiment, we observed an issue during training where the GravMap tokens became entirely NaN, leading to gradient explosion. This, in turn, caused the predicted 6D rotations during inference to become completely disordered. As a result, the converted quaternions $xyzw$ were non-unit quaternions that failed to satisfy $x^2 + y^2 + z^2 + w^2 = 1$. This discrepancy rendered the robotic arm incapable of executing valid rotational actions, leading to a success rate of zero.
>
> We sincerely thank you for highlighting this issue, and we apologize for the lack of clarity in our earlier explanation. ***In the revised manuscript, we have added further clarification on this phenomenon in lines 509–512 (highlighted in blue) to ensure a more thorough understanding***.
>
>
> > It is advisable to conduct real-world experiments, as many related  works, such as Voxposer and 3D Diffuser Actor, also include real-world  demonstrations.
>
> **Response to W4:**  Thank you for the suggestion. ***Based on your comments, we have conducted real-world experiments, and the results are included in Appendix D.5 of the revised manuscript. Additionally, videos of these experiments are available on our project website: https://gravmad.github.io***.

---

> > ### Author Response · Authors · 2024-11-20
> > **Response to Reviewer fAE6 (continued)**
> >
> > > The use of a foundation model, such as GPT-4o, during inference might be time-consuming and could potentially hinder real-time responsiveness in robotic manipulation. Could the authors provide the average inference  time for GravMAD at each sub-task stage?
> >
> > **Response to Q1:** Thank you for the question.
> >
> > |                                                   | Voxposer | Act3D | 3D Diffuser Actor | GravMAD(ours) |
> > | ------------------------------------------------- | -------- | ----- | ----------------- | ------------- |
> > | Avg. Inference Time for Keypose Prediction (secs) | /        | 0.04  | 1.78              | 1.81          |
> > | Avg.  Task Completion Time (secs)                 | 448.01   | 14.08 | 49.45             | 97.04         |
> > | Avg. Inference Time per Sub-task Stage (secs)     | 90.47    | /     | /                 | 40.64         |
> >
> > To address your concern, we measure the inference time of all models under the 8 novel tasks setting using a single NVIDIA 4090 GPU. The results, presented in the table above (in seconds), reveal key trade-offs between inference time and task success rates. Models like Act3D and 3D Diffuser Actor, which do not rely on foundation model inference, exhibit shorter inference times but achieve low success rates. On the other hand, Voxposer spends a considerable amount of time synthesizing trajectories. GravMAD, while requiring more time than Act3D and 3D Diffuser Actor due to the foundation model processing sub-goals, balances inference time with significantly higher task success rates. ***The experimental results and related analysis have been added to Appendix D.1 of the revised manuscript.***
> >
> >
> >
> > > During training, why not generate GravMap in the same way as during  inference? (One possible reason may be that the foundation model’s  inference time is too lengthy?) The GravMap generated from demonstration ground-truths may differ in distribution from the GravMap produced by  the foundation model, which could impact inference performance.
> >
> > **Response to Q2:** Thank you for the question. The use of different methods to generate GravMaps during the training and inference phases is motivated by the following considerations:
> >
> > 1. **Efficiency and reliability during training**: During training, we use Sub-goal Keypose Discovery to extract sub-goals as it is both simple and efficient. While foundation models can generate GravMaps, their outputs are generally coarser, less precise, and slower compared to expert trajectories. For instance, due to limitations such as camera resolution or angles, foundation models may fail to fully observe the scene in certain cases, leading to inaccurate sub-goal positions (as detailed in the failure cases in Appendix B.3). Such inaccuracies compromise the quality of training data. Moreover, using foundation models to process large-scale training data is infeasible due to their slow processing speeds.
> > 2. **Avoiding semantic complexity**: Extracting sub-goals from expert trajectories avoids the complexity of semantic understanding and reasoning. Our approach focuses on analyzing robot actions, as certain actions in expert trajectories inherently carry semantic information (e.g., sub-goals that involve object interactions). These actions often exhibit distinctive features, such as gripper opening and closing. The Keypose Discovery method filters these key actions, reducing the scope for sub-goal selection. This heuristic-based approach is computationally efficient and effective even for long-horizon tasks.
> > 3. **Addressing distribution shift concerns**: The use of different sub-goal generation methods during training and inference may introduce a distributional shift, as sub-goals generated by foundation models during inference are often less precise than those derived from expert trajectories. To mitigate this, we apply data augmentation during training by introducing random offsets to sub-goals derived from expert trajectories, as described in Line 279 of Algorithm 1. These perturbed sub-goals are used to generate GravMaps, reducing the risk of distributional mismatch between training and inference phases.
> >
> > ***Further details about this issue and its mitigation are provided in Appendix C.3 of the revised paper.*** We sincerely thank you for raising this important point, and we hope the additional explanations address your concerns.
> >
> >
> >
> > We appreciate Reviewer fAE6's feedback and will incorporate all suggestions to improve the clarity, completeness, and rigor of the manuscript. Thank you for helping us strengthen our work!

---

> > > ### Author Response · Authors · 2024-11-25
> > > **Response to Reviewer fAE6**
> > >
> > > Thank you once again for your time and effort in reviewing our paper and for providing such valuable feedback! We have carefully addressed the points you raised, including explaining the relationship and differences between the GravMap used in GravMAD and the value maps in Voxposer (**Response to W1**), adding the current limitations of GravMAD's Detector and proposing corresponding solutions (**Response to W2**), clarifying the relationship between quaternions and 6D rotation representations (**Response to W3**), presenting GravMAD's results on 10 real-world robotic manipulation tasks (**Response to W4**), measuring the inference time of all models on the 8 novel tasks (**Response to Q1**), and further elaborating on the reasons for using different methods to generate GravMaps during the training and inference phases (**Response to Q2**). All corresponding clarifications have been incorporated into the revised paper and Appendix, with changes marked in blue.
> > >
> > > As the discussion stage between authors and reviewers is nearing its conclusion, we kindly request the reviewers to review our revised paper and responses. **If our responses have adequately addressed your concerns, we would greatly appreciate it if you could reconsider your scores.**
> > >
> > > If you have any additional questions, we would be more than happy to provide further clarifications. We sincerely look forward to your feedback.

---

### Official Review · Reviewer_2Lap · 2024-11-04

**Soundness:** 2
**Presentation:** 3
**Contribution:** 2
**Rating:** 6
**Confidence:** 5

**Summary:**

The paper proposes a pipeline of Grounded Spatial Value Maps-guided Action Diffusion (*GravMAD*), which is a subgoal-driven, language-conditioned action diffusion framework combining VoxPoser (Huang et al., 2023) with subtask imitation learning based on 3D Diffuser Actor (Ke et al., 2024). The idea is essentially to break tasks into sub-goals based on language instructions, allowing auxiliary guidance to help robots conceptually grasp tasks more accurately than directly using pre-trained foundation models.

The division of subtasks is conducted via Sub-goal Keypose Discovery, which is defined to happen when end effectors make contact or disengage with the objects. The proposed method uses pre-computed keyposes and touch&gripper change detection to segment subtasks for training epochs. During inference, the proposed method uses the LLM-based Planner and Composer from VoxPoser (Huang et al., 2023).

With the segmented subtasks, the proposed method first uses VoxPoser (Huang et al., 2023) to generate 3D value maps *GravMaps* and then uses 3D Diffuser Actor (Ke et al., 2024) to conduct subtask imitation learning and generate action sequences with the condition of tokens from *GravMaps*.

The authors test *GravMAD* on 20 RLBench (James et al., 2020) simulated tasks, including 12 original tasks selected from RLBench and  8 modified new tasks. Empirical results show that the proposed framework outperforms both VoxPoser and 3D Diffuser Actor in most seen and novel tasks, which validates GravMAD's superiority in both generalization and preciseness.

**Strengths:**

1. The paper is well-written, easy to follow, and presents its ideas clearly.
2. The approach of combining VoxPoser's planning and perception ability with precise subtask imitation learning from 3D Diffuser Actor is well-motivated and clever to enhance the model’s long horizon task capability to achieve both VoxPoser's generalization and 3D Diffuser Actor's preciseness.
3. The reported scores are evaluated across 3 seeds, which are statically robust.

**Weaknesses:**

1. The proposed Sub-goal Keypose Discover method is kind of naive. The proposed method simply uses the change in the gripper’s open/close state and the change in touch force to segment a whole task, which could suit simple tasks like moving the chicken but might not work for tasks with longer horizons. This segmenting requires complex reasoning over the concrete task description and comprehensive understanding. During inference, foundation models like VoxPoser can be better at dividing subtasks due to the reasoning ability of LLM. However, for this proposed pipeline where different segmenting methods are used in training and testing, there could even be a domain shift for the subtasks' distribution itself due to unaligned segmenting.
2. The comparison and evaluation are limited. The selection strategy of the 12 tasks over 100 language-conditioned tasks from RLBench is not mentioned. The 8 modified tasks are not justified for their novelty, which might cause a doubt on opportunistic choices.
3. Since both baselines of VoxPoser and 3D Diffuser Actor have conducted real-world evaluations, a real-world comparison for *GravMAD* will be more convincing.

Though conducted in an A+B paradigm on VoxPoser and 3D Diffuser Actor, this work's engineering contribution is solid and the idea of subtask imitation learning is novel. I will raise my score if more validations are provided.

**Questions:**

1. Can you provide more details on the selection strategy of the 12 tasks over 100 language-conditioned tasks from RLBench? Why do you choose 12 and why these 12? The currently designed novel tasks feature scenes and objects similar to those in the training tasks. Can you design more complex novel tasks?
2. Can you evaluate the performance of the proposed method in a real-world experiment? Most setups should be similar to 3D Diffuser Actor and not that hard.

---

> ### Author Response · Authors · 2024-11-20
> **Response to Reviewer 2Lap**
>
> **General Reply:** We sincerely thank Reviewer 2Lap for acknowledging the strengths of our work, including the clarity of our presentation, the effective integration of VoxPoser with 3D Diffuser Actor, and the robustness of our results across multiple experiments. Below, we provide detailed responses to the comments and questions.
>
> > The proposed Sub-goal Keypose Discover method is kind of naive. The  proposed method simply uses the change in the gripper’s open/close state and the change in touch force to segment a whole task, which could suit simple tasks like moving the chicken but might not work for tasks with  longer horizons. This segmenting requires complex reasoning over the  concrete task description and comprehensive understanding. During  inference, foundation models like VoxPoser can be better at dividing  subtasks due to the reasoning ability of LLM. However, for this proposed pipeline where different segmenting methods are used in training and  testing, there could even be a domain shift for the subtasks'  distribution itself due to unaligned segmenting.
>
> **Response to W1:** Thank you for your thoughtful feedback. While the Sub-goal Keypose Discovery method is relatively simple, it is intentionally designed to be practical and effective for robotic manipulation tasks. We appreciate the opportunity to address your concerns in detail:
>
> - **Sub-goal Keypose Discovery**: As described in Section 3.2, our method uses changes in the gripper’s state and contact force to segment sub-tasks efficiently, focusing on critical interactions while minimizing complexity. For example, in the *push buttons* task (Fig. 3 in the main paper), it identifies key sub-goals directly from interaction cues, allowing accurate segmentation without relying on abstract reasoning over task descriptions. This straightforward design aligns well with the requirements of robotic manipulation.
> - **Long-horizon tasks**: Our experiments demonstrate the effectiveness of the method on long-horizon tasks, such as **stack blocks**. As shown in Tables 1 and 2, GravMAD achieved a 35.33% success rate improvement over the best baseline on the basic **stack blocks** task and a 16% improvement on the novel **stack cups blocks** task. These results highlight its robustness and practical applicability.
> - **Domain shift**: We acknowledge that using different sub-goal generation methods during training and inference may introduce distributional shifts. To mitigate this, we apply data augmentation during training by introducing random offsets to sub-goals derived from expert trajectories (Line 279, Algorithm 1). This reduces the risk of distributional mismatch, and ***further details are provided in Appendix C.2 of the revised manuscript***.
>
> We sincerely thank you for your valuable feedback, which has helped us further improve the clarity of the paper. In the future, we plan to enhance the capabilities of Sub-goal Keypose Discovery by incorporating learning-based methods, such as leveraging diffusion models or generative models to generate sub-goals, to tackle more complex robotic manipulation tasks. ***To better address the reviewer’s comments, the above discussion on future improvements has been added to Appendix E of the revised manuscript***.

---

> ### Author Response · Authors · 2024-11-20
> **Response to Reviewer 2Lap (continued)**
>
> > The comparison and evaluation are limited. The selection strategy of the 12 tasks over 100 language-conditioned tasks from RLBench is not  mentioned. The 8 modified tasks are not justified for their novelty,  which might cause a doubt on opportunistic choices.
>
> **Response to W2:** Thank you for your comment. The selection of the 12 base tasks is based on their coverage of the 8 essential action primitives required for robotic manipulation tasks. These tasks are chosen because they provide a solid foundation for learning fundamental skills, enabling generalization to more complex and diverse tasks. ***To clarify the criteria for selecting the base tasks, we add detailed descriptions in Appendix B.1 of the revised manuscript, highlighted in blue in Figure 7 and lines 1017–1024.***
>
> We also appreciate your feedback regarding the definition of the 8 modified novel tasks. To address your concerns, we have added detailed definitions of these novel tasks and further categorized their novelty. These 8 novel tasks introduce three distinct types of challenges:
>
> 1. **Action Understanding** (*meat on grill*, *close drawer*): Tasks involving changes in interaction actions with objects.
> 2. **Visual Understanding & Language Reasoning** (*stack cups blocks*, *push buttons light*, *close jar banana*, *condition block*): Tasks requiring novel operational rules or conditions compared to known tasks, including two long-horizon tasks (*stack cups blocks* and *condition block*).
> 3. **Robustness to Distractors or Shape Variations** (*stack cups blocks*, *push buttons light*, *close jar banana*, *close jar distractor*, *open small drawer*, *condition block*): Tasks requiring interaction based on fixed object attributes such as color, size, distance, or the presence of distractors.
>
> **These categories and definitions have been added to Appendix B.2 of the revised manuscript, highlighted in blue, specifically in Figure 10 and lines 1198–1209.** This additional explanation aims to better illustrate how GravMAD leverages these challenges to demonstrate improved generalization capabilities.
>
> > Can you provide more details on the selection strategy of the 12 tasks  over 100 language-conditioned tasks from RLBench? Why do you choose 12  and why these 12? The currently designed novel tasks feature scenes and  objects similar to those in the training tasks. Can you design more  complex novel tasks?
>
> **Response to Q1:** Thank you for the suggestions. The selection criteria and design rationale for the 12 base tasks and 8 novel tasks are detailed in our **Response to W2** above. Following the reviewer’s suggestion, we have designed 3 additional novel tasks and conducted testing with GravMAD and baseline methods on these tasks. ***The results of these additional experiments have been included in Appendix D.2 of the revised manuscript.***
>
> > Since both baselines of VoxPoser and 3D Diffuser Actor have conducted real-world evaluations, a real-world comparison for *GravMAD* will be more convincing.
> >
> > Can you evaluate the performance of the proposed method in a real-world  experiment? Most setups should be similar to 3D Diffuser Actor and not  that hard.
>
> **Response to  W3 &Q2:** Thank you for the suggestion regarding real-world experiments. ***In response, we have included the results of GravMAD on 10 real-world robotic manipulation tasks in Appendix D.5 of the revised manuscript. Additionally, videos showcasing these experiments have been made available on our anonymous project website: [https://gravmad.github.io](https://gravmad.github.io).***
>
> We appreciate Reviewer 2Lap's constructive comments and have incorporated all suggestions to improve the clarity, completeness, and rigor of the manuscript. Thank you for helping us strengthen our work!

---

> > ### Author Response · Authors · 2024-11-25
> > **Response to Reviewer 2Lap**
> >
> > Thank you once again for your time and effort in reviewing our paper and for providing such valuable feedback! We have carefully addressed the points you raised, focusing on clarifying our Sub-goal Keypose Discovery method (**Response to W1**), refining the selection criteria for base and novel tasks (**Response to W2**), designing and experimentally testing three additional challenging novel tasks (**Response to Q1**), and presenting GravMAD's results on 10 real-world robotic manipulation tasks (**Response to W3 & Q2**). All corresponding clarifications have been incorporated into the revised paper and Appendix, with changes marked in blue.
> >
> > As the discussion stage approaches its conclusion, we kindly request the reviewers to review our revised paper and responses. **If our responses have adequately addressed your concerns, we would greatly appreciate it if you could reconsider your scores.**
> >
> > If you have any additional questions, we are more than happy to provide further clarifications. We sincerely look forward to your feedback.

---

> > > ### Author Response · Authors · 2024-11-27
> > > **Response to Reviewer 2Lap**
> > >
> > > We would like to express our sincere gratitude to you once again for your valuable feedback, which has played a crucial role in improving our manuscript and enhancing its competitiveness.
> > >
> > > **Although the discussion period has been extended, we would like to kindly remind you that the deadline for authors to upload the revised PDF is fast approaching**. If you have any remaining questions or concerns, we would greatly appreciate your prompt response. If you believe our responses have addressed your concerns, we would be grateful if you could consider revising our scores accordingly. We are happy to address any outstanding issues and are willing to make further revisions based on your feedback in the final stages!
> > >
> > > Thank you once again for your dedicated service to the academic community.

---

> ### Author Response · Authors · 2024-12-01
> **A Kind Reminder [Deadline Approaching]**
>
> Dear Reviewer 2Lap,
>
> We would like to thank you once again for taking the time to review our work and provide feedback on our manuscript. As the extended discussion deadline (December 2) approaches, this is a gentle reminder to let us know if we have satisfactorily addressed the reviewer's concerns, focusing on: clarifying our Sub-goal Keypose Discovery method (Response to W1), refining the selection criteria for base and novel tasks (Response to W2), designing and experimentally testing three additional challenging novel tasks (Response to Q1), and presenting GravMAD's results on 10 real-robot manipulation tasks (Response to W3 and Q2). If you find it appropriate, we would be grateful if you could update your scores. We are happy to address any additional remaining concerns. We are grateful for your service to the community.
>
> Regards,
>
> The Authors of Submission 3039

---

> > ### Comment · Reviewer_2Lap · 2024-12-02
> >
> > Thanks for clarifying the concerns. I'd like to raise my scores.

---

> > > ### Author Response · Authors · 2024-12-02
> > >
> > > We are delighted to hear that our response and the revisions to the manuscript have addressed your concerns, and we greatly appreciate your decision to raise your score. Thank you once again for your time and valuable feedback!

---

### Author Response · Authors · 2024-11-20
**General response to all reviewers**

We sincerely thank all reviewers for their constructive feedback, comments, and suggestions.

We summarize our modifications according to feedback:

- Main paper
    - Abstract: Modified vague terms such as "3D tasks" and added descriptions of real-world robotic tasks.
    - Sec 1: Revised all unclear statements pointed out by **Reviewer vbZp** and added descriptions of the real-world robotic experiments.
    - Sec 2: Revised all unclear statements pointed out by **Reviewer vbZp**
    - Sec 3.1: Further explain why the 6D rotation representation is used.
    - Sec 3.3: More details on how the GravMap token is utilized within the network.
    - Sec 4.1: Further clarify the boundaries between the 12 base tasks and 8 novel tasks in the experiments, and add descriptions of the real-world robotic experiments.
    - Sec 4.4：Further explain the ablation results of w/o GravMap and w/o Cost Map.
    - Sec 5：Add descriptions of the real-world robotic tasks.
- Appendix
    - Sec B: Further clarify the capability boundaries between the 12 base tasks and the 8 novel tasks, as well as the reasons for selecting them.
    - Sec C.1: The Relationship and Differences Between GravMap and Voxposer.
    - Sec C.2: The reason for not using the rotation map from Voxposer
    - Sec C.3: Further Details on Sub-goal Keypose Discovery
    - Sec D.1: Additional Experimental Results on Inference Time
    - Sec D.2: Additional Baseline Experiments
    - Sec D.3: Additional Experimental Results on Scalability of GravMAD
    - Sec D.4: Additional Novel Tasks
    - Sec D.5: Real World Evaluation
    - Sec E: Limitations and Potential Solutions
- Anonymous project website https://gravmad.github.io: Add video demonstrations of real-world robotic tasks.

If reviewers have any further questions, please feel free to ask—we would be very happy to discuss them！

---

> ### Author Response · Authors · 2024-11-22
> **General response to all reviewers (continued)**
>
> We thank the reviewers (Reviewer 2Lap, Reviewer fAE6, Reviewer SKBE, Reviewer vbZp) for their valuable feedback.
>
> We are encouraged that the reviewers recognized **the strong and clear research motivation behind our GravMAD framework** (Reviewer 2Lap, Reviewer SKBE, Reviewer vbZp) and acknowledged that **combining imitation learning with foundation models' generalization capabilities to tackle 3D robotic manipulation tasks is both challenging and meaningful** (Reviewer vbZp). The reviewers also highlighted **our comprehensive experimental design, robust results, and strong generalization capabilities** (Reviewer 2Lap, Reviewer SKBE, Reviewer vbZp). Moreover, they agreed that **our paper is well-structured, clearly explained, and enhanced by effective visualizations** (Reviewer 2Lap, Reviewer fAE6, Reviewer SKBE, Reviewer vbZp). Lastly, **the ablation studies were commended for providing valuable insights into the contributions of key components** (Reviewer 2Lap, Reviewer vbZp).
>
> We have addressed all the reviewers' comments below and incorporated the corresponding revisions into the revised manuscript. The key updates in the revised version have been summarized in the previous general response, with the modified sections highlighted in BLUE.
>
> **One specific point we would like to reiterate is the "w/o Cost map" ablation experiment.** We are currently conducting additional training experiments based on Reviewer vbZp's suggestion to address the zero-gradient issue. We expect the results to be available toward the later stages of the rebuttal phase. If this modification successfully resolves the zero-gradient issue, we will include the updated experimental results in the revised manuscript. Conversely, if the issue persists, we will remove this ablation experiment, as suggested by Reviewer vbZp, to ensure the clarity and scientific rigor of the paper.
>
> We sincerely thank the reviewers for their constructive feedback and the valuable time devoted to our work. If our responses adequately address the concerns, we kindly request the reviewers to consider raising the rating and supporting us for acceptance.

---

> > ### Author Response · Authors · 2024-11-25
> > **General response to all reviewers (continued)**
> >
> > We sincerely thank all the reviewers once again for their valuable comments and suggestions!
> >
> > **Regarding the "w/o Cost map" ablation, we have completed new experiments and incorporated the results into Appendix D.5 of the revised manuscript. Additionally, we have further refined the description of the "w/o Cost map" performance in Section 4.4 of the main paper and included a reference to Appendix D.5.** The new experimental results show that changing the gripper map's closed state representation from 0 to -1 in the "w/o Cost map" setting allows the encoder to correctly process this structure. However, removing the cost map results in a significant performance drop compared to the original model: an 11.97% decrease in testing performance on 12 base tasks and a 21.04% decrease on 8 novel tasks. These findings clearly demonstrate the critical role of the cost map in ensuring the performance of the GravMAD model.
> >
> > As the discussion stage between the authors and reviewers approaches its conclusion, we kindly request the reviewers to review our revised paper and responses. **If our responses have adequately addressed your concerns, we would greatly appreciate it if you could reconsider your scores.**
> >
> > If the reviewers have any additional questions, we would be more than happy to provide further clarifications. We sincerely look forward to your feedback.

---

### Meta-Review · Area_Chair_uZtF · 2024-12-21

**Metareview:**

The paper presents a framework for robotic manipulation that combines prior work on action diffusion and robotic control with foundation models. The approach demonstrates solid generalization capabilities and outperforms state-of-the-art baselines.

Strengths:

- Clear motivation and robust experimental design.

- Notable performance improvements on training and unseen tasks.

- Extensive ablation studies and real-world validations.

Weaknesses:

- Limited novelty beyond combining existing frameworks.

- Challenges with robustness and precision of certain components.

- Experiments are limited to RLBench, raising concerns about broader applicability, which was partially addressed by the authors with additional real-world experiments in the rebuttal.

While the paper has areas that could be improved, such as enhancing methodological contributions and broadening experimental validations, it provides useful insights into robotic manipulation. The authors have addressed most of the reviewers' concerns through detailed rebuttals and revisions, including real-world experiments and clarifications. Therefore, I recommend acceptance.

**Additional Comments On Reviewer Discussion:**

During the discussion, reviewers raised concerns about GravMAD's limited novelty, the simplicity of its Sub-goal Keypose Discovery method, evaluation confined to RLBench, and technical issues like quaternion representation and inference speed. The authors addressed these by mitigating domain shifts through data augmentation, and expanding evaluation with real-world experiments and additional metrics. They clarified technical issues and provided future improvement plans. These efforts resolved most concerns, leading to an overall positive impression and a decision to accept the paper as above the threshold.

---

### Decision · Program_Chairs · 2025-01-22

Accept (Poster)